# Modelling flood frequency and magnitude in a glacially conditioned, heterogeneous landscape: testing the importance of land cover and land use.

Pamela E. Tetford[1], Joseph R. Desloges[1]

5    Department of Geography, University of Toronto, Toronto, ON, M5S 3G3, Canada

*Correspondence to*: Pamela E. Tetford (pam.tetford@mail.utoronto.ca)

**Abstract.** A reliable flood frequency analysis (FFA) requires selection of an appropriate statistical distribution to model historic streamflow data and, where streamflow data are not available (ungauged sites), a regression-based regional flood frequency analysis (RFFA) often correlates well with downstream channel discharge to drainage area relations. However, the predictive strength of the accepted RFFA relies on an assumption of homogeneous watershed conditions. For glacially conditioned fluvial systems, inherited glacial landforms, sediments, and variable land use can alter flow paths and modify flow regimes. This study compares a multi-variate RFFA that considers 28 explanatory variables to characterize variable watershed conditions (i.e., surficial geology, climate, topography, and land use) to an accepted power-law relationship between discharge and drainage area. Archived gauge data from southern Ontario, Canada are used to test these ideas. Mathematical goodness-of-fit criteria best estimate flood discharge for a broad range of flood recurrence intervals, i.e., 1.25, 2, 5, 10, 25, 50, and 100 years. The Log-Normal, Gumbel, Log-Pearson Type III, and Generalized Extreme Value distributions are found most appropriate in 42.5%, 31.9%, 21.7%, and 3.9% of cases, respectively, suggesting that systematic model selection criterion is required for FFA in heterogeneous landscapes. Multi-variate regression of estimated flood quantiles with backward elimination of explanatory variables using principal component and discriminant analyses reveal that precipitation provides a greater predictive relationship for more frequent flood events, whereas surficial geology demonstrates more predictive ability for high magnitude, less frequent flood events. In this study, all seven flood quantiles identify a statistically significant two-predictor model that incorporates upstream drainage area and the percentage of naturalized landscape with 5% improvement in predictive power over the commonly used single-variable drainage area model ($p < 2.2e\text{-}16$). Leave-one-out model testing and an analysis of variance (ANOVA) further support the parsimonious two-predictor model when estimating flood discharge in this low-relief landscape with pronounced glacial legacy effects and heterogenous land use.

**Keywords:** flood frequency, FFA, RFFA, multi-variate modelling, land use, glacial conditioning

## 1. Introduction

A reliable assessment of flood frequency and flood magnitude over space and time is critical for urban planning and infrastructure engineering that depends on flood probability (Basso et al., 2016). Flood magnitude, frequency, and duration are primary drivers of channel erosion and stream morphology (Taniguchi & Biggs, 2015) as a self-shaping alluvial channel entrains and transports sediment to adjust its dimensions, planform pattern, bed characteristics, and gradient in response to varying flow levels (Church & Ferguson, 2015). So reliable estimates of flood frequency are important for understanding geomorphic channel change.

A regional flood frequency analysis (RFFA) can be very important in determining the probability of extreme flood events where streamflow data are not readily available (Ahn & Palmer, 2016) by transferring observed hydrologic information from a group of gauged sites to comparative ungauged sites as a representation of flow statistics using hydrological variables (Odry & Arnaud, 2017). A common approach to RFFA consolidates data samples from many measuring sites and uses ordinary least-squares (OLS) regression to identify a relationship between mean annual floods of multiple basins and some basin characteristic (e.g., drainage area). As the source area for channel discharge, drainage area is a widely used proxy for channel discharge (Galster, 2006; Knighton, 1999). It has become an accepted practice to model discharge using a single-variable power-law relationship between discharge ($Q$) and drainage area ($A_d$) of the form

$$Q = \alpha A_d^{\beta} \tag{1}$$

where $A_d$ is the upstream drainage area and the coefficient $\alpha$ and exponent $\beta$ are empirically derived by statistical regression (Dunne & Leopold, 1978; Knighton, 1999; Phillips & Desloges, 2014). This scaling relationship can be rewritten as

$$\log Q = \log \alpha + \beta \log A_d . \tag{2}$$

The reliability of this single-variable predictive relationship, however, relies on the relative regional homogeneity of the landscape, with similar basin conditions and climate (Ahn & Palmer, 2016; Hosking & Wallis, 1993; Phillips & Desloges, 2014).

To estimate how often a specified flood event (or channel discharge) will occur, flood frequency analysis (FFA) is widely used (Farooq et al., 2018). Most often, an FFA uses the occurrence of extreme flood events to estimate the return period, $T$, of flood quantiles, $Q(T)$, using a fixed probability model based on long-term, historic flow data from a gauge station (Di Baldassarre et al., 2009). This probabilistic approach "fits" the site-specific data to a statistical distribution to predict the likelihood of future flood events. To provide flexibility of fit, statistical probability distributions require two to four parameters (Zhang et al., 2020). The choice of the probabilistic model that best represents the observed data and the estimation of a distribution's parameters affects the reliability of flood prediction (Cunnane, 1973; Farooq et al., 2018; Laio et al., 2009). Poor model application and fit can lead to unreliable estimates (Basso et al., 2016). The Generalized Extreme Value (GEV) distribution, Gumbel Maximum or Extreme Value Type I (EV1) distribution, Log-Normal (LN) distribution, and Log-Pearson Type III

(LP3) distribution have traditionally been recommended to characterize flood probability based on goodness-of-fit (Onen &
Bagatur, 2017; Laio et al., 2009). The LP3 and GEV distributions use three parameters, i.e., location, scale, and shape, and the
EV1 and LN distributions use two parameters, i.e., location and scale, to fit data distributions (see Appendix A). In Canada, it
is recommended that FFA studies draw from the Normal, GEV, and Pearson distribution families. Distribution fitting with
more than three parameters is not recommended due to the limited record lengths of Canadian gauge stations (Natural
Resources Canada, 2019). For regions with diverse flood characteristics, multiple distributions may apply for different
catchments requiring site specific selections (Zhang et al., 2020). Since 1967, Guidelines for Determining Flood Flow
Frequency, Bulletin 17C of the U.S. Geological Survey (USGS) recommend the use of the LP3 distribution as an appropriate
statistical distribution to characterize the probabilies of annual flood series (USGS, 2019) but the recent HEC-SSP Statistical
Software Package Version 2.2 includes the ability to perform two goodness-of-fit tests for up to 19 statistical distributions
(Hydrologic Engineering Center, 2019). Recent research indicates that estimation of flood frequency and magnitude improves
with the application of a systematic and objective model selection criteria when fitting observed flow data to a statistical
probabilistic curve (Di Baldassarre et al., 2009).

Research suggests that the spatial variability of basin attributes (i.e., topographic relief, climate, vegetation, and land use) and
subsurface characteristics which influence hydrological and fluvial function are controlling factors of a fluvial system's
drainage efficiency and are relevant to the flow response in a catchment (Di Lazzaro et al., 2015; Fryirs & Brierley, 2012;
Galster, 2006; Oudin et al., 2008). Additionally, landscape modifications that decrease infiltration will impose changes to river
hydrology (Ashmore, 2015; Ghunowa et al., 2021; Taniguchi & Biggs, 2015; Winter, 2001) with a downstream cascading
effect on flow regime (Royall, 2013). Human occupation, landscape manipulation, and the generation of impervious surfaces
associated with urbanization have the most profound impact on hydrogeomorphic responses, particularly in smaller watersheds
(Pasternack, 2013; Royall, 2013). And a fluvial system's response to human-induced land use change (or its sensitivity to
change) will vary, depending on basin attributes (i.e., configuration, geomorphology, and sediment retention) (Royall, 2013).
For this reason, the spatial heterogeneity across a landscape will likely produce a variation in flood response that may best be
captured using a multi-variate RFFA approach that considers parameterization of relevant basin characteristics (i.e.,
topographic relief, land use, vegetation, and subsurface geology) as a set of explanatory variables to estimate flood discharge
(Ahn & Palmer, 2016).

Recent works have highlighted the impact of geomorphic spatial heterogeneity on the basin hydrologic response (Ahn &
Palmer, 2016; Di Lazzaro et al., 2015; Taniguchi & Biggs, 2015). However, many rapid geomorphic studies have relied on
just catchment area as the leading attribute for estimating channel forming discharge (Ashmore et al., 2022). This study seeks
to explore additional explanatory hydrologic and land use controls that improve the predictive strength of this relationship in
a heterogeneous landscape. This multi-variate approach uses exploratory statistical analysis to better understand the link
between intra-catchment variability and hydrological function. In this study:

1) An FFA is completed to model reliable estimations of discharge for a broad range of flood recurrence intervals (i.e., $Q_{1.25}$, $Q_2$, $Q_5$, $Q_{10}$, $Q_{25}$, $Q_{50}$, and $Q_{100}$). Model selection is determined by applying systematic and objective model selection criteria to optimize model fit to long term site-specific flow data (>10 years). A test sample of 207 individual gauge sites within a glacially conditioned regional setting is used.

2) The widely used single-variable RFFA (Eq. (1)) is derived to characterize the relationship between discharge ($Q$) and site-specific drainage area ($A_d$) for the test region using optimized estimates of a broad range of flood quantiles to test the predictive power of a single hydrologic variable in a spatially heterogeneous landscape.

3) A multi-variate, regression based RFFA is presented that considers the spatially variability of hydrologic controls in the context of inherited glacial landforms, sediments, and land use. To achieve this goal, twenty-eight (28) predictor variables are explored representing basin characteristics (i.e., topographic relief, climate, land use, vegetation, and subsurface geology). A backward elimination approach is employed (i.e., discriminant and principal component analyses, regression diagnostics) to identify the most parsimonious discharge models for recurrence intervals of 1.25, 2, 5, 10, 25, 50, and 100 years.

4) The predictive power of a multi-variate derived RFFA that considers multiple basin hydrologic controls is compared to a generally accepted single-variable RFFA in a spatially heterogeneous setting.

## 2. Regional Setting

This flood frequency study focuses on a test region of peninsular southern Ontario, Canada (Figure 1) that is bounded by the Canadian Shield to the north, the three lower Great Lakes, Huron, Erie, and Ontario to the southwest and the Ottawa and St. Lawrence Rivers to the east. Located within the North American Great Lakes watershed, it is a region of modest relief, with elevation ranging from 544 m asl near Lake Huron draining by way of the St. Lawrence River lowlands at less than 70 m to the Atlantic Ocean (Larson & Schaetzl, 2001). Convective, synoptic, and tropical systems that influence the humid, continental climate of the region are enhanced by local, regional, and topographic conditions (Paixao et al., 2011). Moisture and temperature associated with the Great Lakes influence inland precipitation for up to 50 km. Consequently, the mean annual precipitation varies regionally from 800 mm to 1200 mm (Paixao et al., 2011). During winter months, precipitation typically accumulates in the form of snow, generating spring snowmelt floods that dominate river flow regimes (Javelle et al., 2003). The surficial geology of the region, and the hydrologic controls exerted by the parent materials, are the product of the region's glacial history (Chapman & Putnam, 1984). Recurring continental glaciations over the last ~2 million years have topographically influenced the fluvial drainage networks of southern Ontario (Desloges et al., 2020; Fulton et al., 1986).

Deglaciation, approximately 12 to 13 thousand years ago, has left pronounced glacial legacy effects with complex sequences of subglacial, ice-contact, and proglacial sediments deposited during the final retreat of the Laurentide Ice Sheet (Larson & Schaetzl, 2001; Phillips & Desloges, 2014, 2015). The most common physiographic features include sheets of till, finer glaciolacustrine plains of sand or clay, glaciofluvial outwash deposits of sand, gravel, silts and clays, and a configuration of moraines (Thayer et al., 2016). Two significant post-glacial geomorphic features are the Niagara Escarpment and the Oak

Ridges Moraine (Figure 1). The Niagara Escarpment is a Paleozoic limestone bedrock ridge resulting from differential glacial

erosion and weathering of harder and softer rock that arches from the region between Lakes Ontario and Erie, bypassing Lake

Ontario and extending northward to Georgian Bay (Chapman & Putnam, 1984; Phillips & Desloges, 2014).

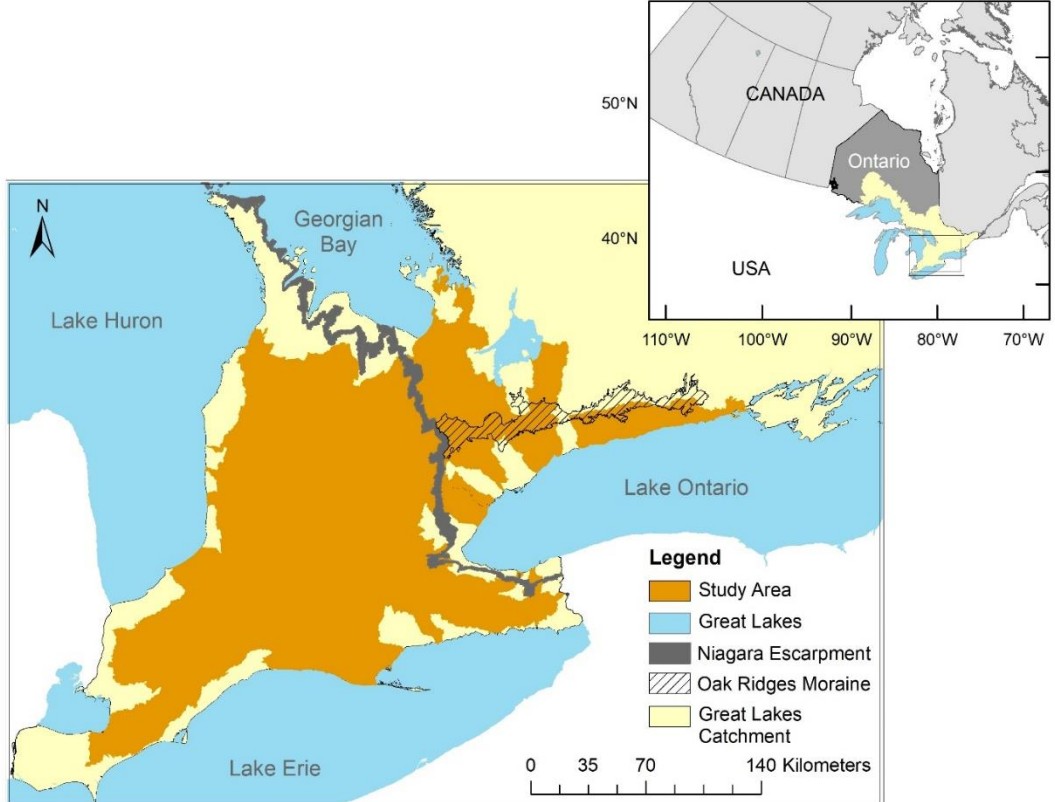

**Figure 1 - Map identifying the study area (indicated in orange) and two significant post-glacial geomorphic features that influence drainage networks of southern Ontario (i.e., the Niagara Escarpment and the Oak Ridges Moraine). The inset map (upper right)**
**indicates the study region within the Ontario portion of the Laurentian Great Lakes catchment relative to Canada.**

Several preglacial rivers have carved deep valleys into the Niagara Escarpment; however, Late Pleistocene glaciations have

infilled these valleys with varying thicknesses of till (Chapman & Putnam, 1984) directing catchment flow mostly away from

the escarpment crest. The Oak Ridges Moraine is a stratified kame moraine of glacial drift that extends from the Niagara

Escarpment 160 km eastward across south-central Ontario (Phillips & Desloges, 2014). This massive ridge forms a drainage

divide, separating catchments flowing north to Georgian Bay/Lake Huron and south to Lake Ontario. Glacial sediments

typically blanket the study area at a thickness of 50 m, and up to 350 m in some places (Larson & Schaetzl, 2001). In many

areas, where stratified limestones and shales of the Palaeozoic age lie beneath the thick glacial overburden, fertile soils rich in

calcium carbonate and clay are produced (Desloges et al., 2020; Phillips & Desloges, 2014, 2015). These fertile soils support

southern Ontario's widespread agricultural development (Donnan, 2008).

More recent European settlement and regional expansion have resulted in differentiated land use with extensive agricultural land, natural and reforested areas, and clustered urban settlement (Chapman & Putnam, 1984). The southern Ontario region continues to accommodate an increasing population. Drawn by employment, most settle in built-up cities and surrounding areas, driving clustered regional urbanization that consumes surrounding rural lands. However, a comparable demand to expand the total area of cropland has also occurred to support larger farming operations (Donnan, 2008).

## 3. Methods and Data Collection

An overview of the methodology for this study is provided in Figure 2.

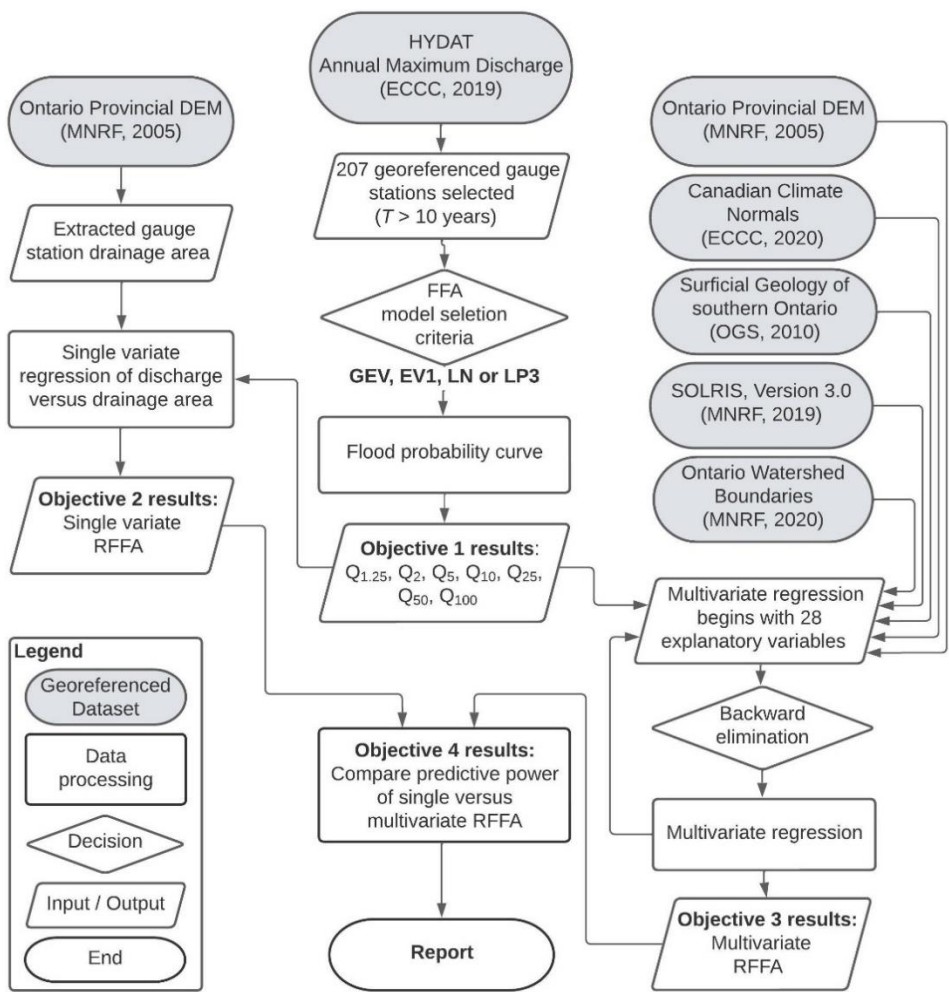

**Figure 2 - Flow chart of FFA and comparison of a multi-variate RFFA to a single-variable RFFA that uses a discharge to drainage area relationship. The regression-based multi-variate RFFA employs sub-basin characterization and backward elimination of**
**explanatory variables to determine the most parsimonious model to predict discharge over seven (7) flood quantiles.**

## 3.1 Estimation of flood frequency

A Station Meta Data Index for 1188 Ontario georeferenced stream gauges from the HYDAT database of the Water Survey of Canada (WSC) monitoring program is accessed online at https://wateroffice.ec.gc.ca/mainmenu/historical_data_index_e.html (Environment and Climate Change Canada (ECCC), 2019). The quality of gauge data depends on the selected measurement techniques, computation methods, and physical conditions at the monitoring sites (i.e., ice and other influences). However, the WSC performs regular audits of field activities and adheres to standard operating procedures to improve data quality (ECCC, 2019). Gauge locations are sorted by catchment and synthesized to identify gauges specific to the southern Ontario region.

Retention of station data is based on three criteria: (1) the gauge station lies within the peninsular region of southern Ontario (2) the gauge station exists for a fluvial system with known field survey data (i.e., Annable (1995, 1996) and Phillips (2014)), and (3) streamflow data represent a minimum of 10 years of operation, continuous (non-seasonal) year-round operation. These criteria yield 207 gauge stations from the HYDAT database with a minimum operation period of 10 years, an average of 42.5 years (+/-1.7 years, median=42 years), and a maximum operation period of 106 years. Two approaches are commonly used for FFA: the annual maxima series (AMS) and the partial duration series (PDS). The AMS approach uses the highest annual discharge from the recorded mean daily discharge values at a gauge, ensuring statistical independence of observations between years (USGS, 2019). Although some research suggests that the instantaneous maximum discharge may command greater geomorphic significance, the mean daily discharges provide a larger dataset with fewer gaps in the discharge records. Alternatively, the PDS approach (or Peak-over-Threshold) uses floods that exceed some base threshold discharge ($q_0$), regardless of the time distribution (USGS, 2019). The AMS approach has been shown to be more efficient than the PDS approach for floods $Q(T)$ when $T>10$ (Cunnane, 1973). This study uses the AMS of mean daily discharge (m$^3$/s) for flood recurrence computations at each gauge station. The MSClaio2008 R function, part of the package nsRFA in R, is used to compare the LP3, EV1, GEV, and LN distributions to the annual maximum discharge data. No prior processing is implemented to fit the distributions. Each flood dataset is fit to each of the four candidate models (i.e., GEV, EV1, LN, and LP3) in the form of probability distributions with parameters estimated using the maximum likelihood method. Model selection criteria, including the Akaike Information Criterion (AIC), the Bayesian Information Criterion (BIC), the Anderson-Darling Criterion (ADC), and the corrected Akaike Information Criterion (AICc) where the sample size, n, is small with respect to the number of estimated parameters, p, such that n/p<40, are separately applied to each candidate model to evaluate the model which most closely fits the flood data, similar to the methods of others (Baldassarre et al., 2009; Farooq et al., 2018; Laio et al., 2009). These model selection criteria are shown to provide good operational strategy when applied to frequency analysis of hydrological extremes by enabling a systematic and objective mathematical test of model fit (Laio et al., 2009). For each flood dataset, the MSClaio2008 function returns the distribution that is most often selected by the selection criteria. The selected optimal distribution for each gauge dataset is used to model flood recurrence using the cumulative probability. Flood quantiles for seven recurrence intervals (RIs) of 1.25, 2, 5, 10, 25, 50, and 100 years (i.e., $Q_{1.25}$, $Q_2$, $Q_5$, $Q_{10}$, $Q_{25}$, $Q_{50}$, $Q_{100}$) are derived directly from individual gauge data and, therefore, reflect the upstream conditions of the corresponding drainage basin.

### 3.2 Selection of explanatory variables

**3.2.1 Gauge station drainage area**

Catchment basins of the study area are delineated based on the hierarchical framework of the Ontario Watershed Boundaries, published by the Ontario Ministry of Natural Resources and Forestry (OMNRF) (2020). The digital geospatial datasets are accurate to within 100 m and accessed online from https://data.ontario.ca/dataset/ontario-watershed-boundaries . Basins are first identified according to Tertiary level watersheds.

The site specific drainage area for each gauge station is evaluated based on Ontario's hydrologically enforced Provincial Digital Elevation Model (DEM) – Version 2.0.0 of the Ontario Ministry of Natural Resources (OMNR) (2005) following Phillips & Desloges (2014). Hydrological enforcement ensures that drainage occurs in a down-slope direction, facilitating the construction of a flow accumulation raster necessary to establish the upstream drainage area of each gauge station.

**3.2.2 Sub-basin attributes**

To characterize the upstream hydrologic, geomorphic, and land use conditions affecting channel discharge, the 16 tertiary level catchments of southern Ontario are subdivided into 45 sub-watersheds demarcated by quaternary level boundaries. Georeferenced gauge stations are clustered within sub-basin units to best represent the immediate upstream hydrological conditions. Sub-basin attributes are selected to characterize the drainage area conditions for each gauge station. Using digital cell counts and zonal statistics from multiple sources, sub-basin characteristic variables are extracted from four geospatial

raster datasets to represent topography, land use, precipitation, and hydrologic properties from a geomorphic perspective:

a) Ontario's Provincial DEM, Version 2.0.0, a hydrologically enforced tiled raster dataset with a cell resolution of 10 m and vertical accuracy of 5 m.

b) the southern Ontario Land Resource Information System (SOLRIS) Version 3.0, accessed online at https://geohub.lio.gov.on.ca/documents/lio::southern-ontario-land-resource-information-system-solris-3-0/about , a

comprehensive, digital landscape level inventory published by the OMNRF(2019) that identifies urban, rural, and natural features at a 15 m resolution derived from Landsat-8 OLI imagery acquired from 2014 to 2017.

c) the Canadian Climate Normals 1981-2010 accessed at https://climate.weather.gc.ca/climate_normals/ . The Canadian Climate Normals are commonly used to assess regional climate and adhere to the accepted standards of the World Meteorological Organization which recommends 30-year records to eliminate year to year variation (ECCC), 2020).

d) the revised Surficial Geology of Southern Ontario (MRD 128–Revised), accessed online at http://www.geologyontario.mndm.gov.on.ca/mndmaccess/mndm_dir.asp?type=pub&id=MRD128-REV which provides a seamless, standardized map of the geology, primary material, genesis, and formation coverages for southern Ontario (Ontario Geological Survey (OGS), 2010).

For each predictor variable (i.e., attribute), a single output value is produced and applied to the gauge(s) within a sub-basin. Since many sub-basins contain more than one gauge station, some gauges share the same topographic, land use, climate, and geomorphic values but possess their own unique drainage area value. All mapping and spatial analyses use a combination of standard GIS software and spreadsheet algorithms. Maps are projected to the Universal Transverse Mercator (UTM, Zone 17N), referenced to the North American Datum (NAD1983).

### 3.3 Comparison of single-variable RFFA to multi-variate RFFA

For each of the seven flood quantiles, a single-variable relationship (Eq. (1)) between discharge and drainage area is obtained by statistical regression. Multi-variate relationships between the explanatory variables and each of the quantile discharge datasets are assessed by applying OLS regression. OLS assumes that the set of explanatory variables (i.e., basin characteristics) and errors must be independent to avoid bias. When characterizing natural systems, the potential exists for some variables to correlate with other variables due to their representation of related natural phenomena, often indicated by high correlations between variables suggesting a duplication of information captured (Ahn & Palmer, 2016; Phillips & Desloges, 2015). To identify the most parsimonious discharge model for RIs of 1.25, 2, 5, 10, 25, 50, and 100 years, regression models are developed using a backward elimination strategy (Figure 3):

1) discriminant analysis, similar to Ahn & Palmer (2016), tests for variable independence and identifies highly correlated variables. A principal components analysis (PCA) explores the most important influences on channel discharge. PCA has been shown to be an effective tool for variable reduction that provides a statistical basis to discard redundant variables (King & Jackson, 1999). A simple Pearson correlation is applied to all predictor variables (criterion $|r|>0.6$). Preference is granted to predictors with a stronger theoretical association to channel discharge. Similarly, Ahn and Palmer (2016) apply a criterion of $|r|>0.7$ to eliminate strongly correlated variables,

2) an iterative process of multi-variate regression diagnostics is applied, following others (Roman et al., 2012; Sheather & Oostrom, 2009), to remove variables that demonstrate little or no predictive power, and

3) models are evaluated for performance using an analysis of variance (ANOVA) that compares the residual sum of squares of the multi-variate model to the single-variate model, and leave-one-out cross validation (LOOCV) to assess the predictive capabilities of the model in practice.

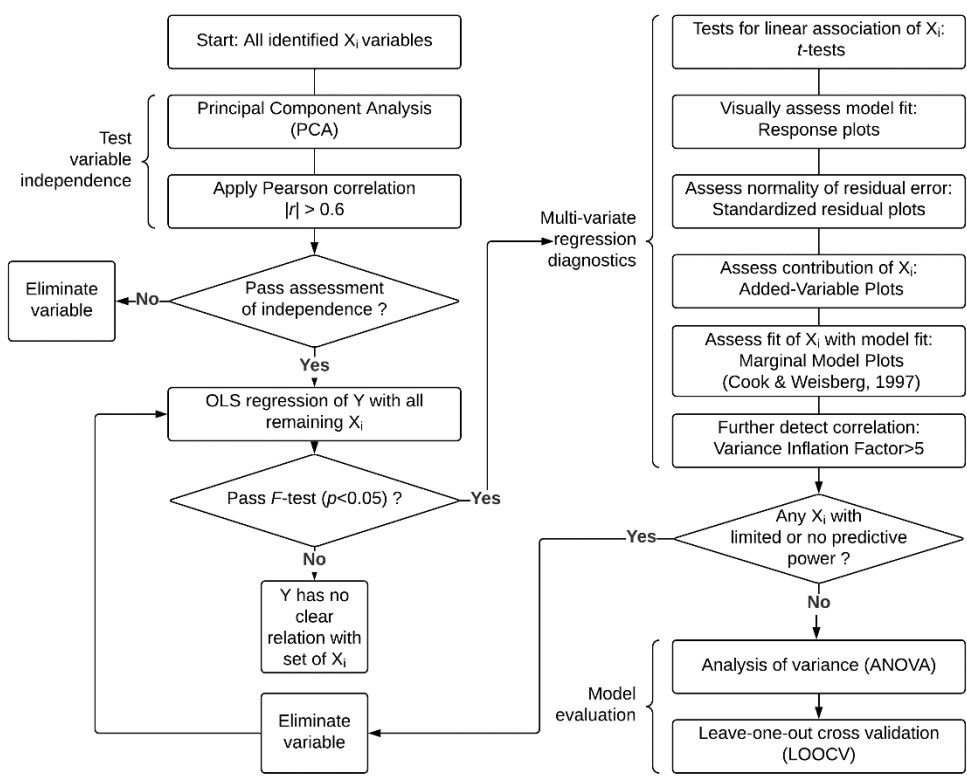

**Figure 3 – Flowchart of discriminant analysis and backward elimination strategy employing tests for variable independence, multi-variate regression diagnostics, and model evaluation.**

## 4. Regression Model Inputs

### 4.1 Flood quantiles as dependent variables

The model selection criteria determines that 42.5% of the 207 hydrometric gauge records are most suited to an LN distribution,

31.9% to an EV1 distribution, 21.7% to an LP3 distribution, and 3.9% to a GEV distribution (Figure 4). Goodness-of-fit tests suggest that all four distributions are potentially suitable for modelling flood extremes from gauges in southern Ontario. For 74.4% of the gauge records tested, the selection criteria chose a 2-parameter model (i.e., LN or EV1) over a 3-parameter model (i.e., LP3 or GEV). The 2-parameter EV1 model is found to be five (5) times more likely to be selected as the optimal distribution over its 3-parameter parent model, GEV. The GEV distribution is only selected in a limited number of cases. In

general, there is no single "best fit" distribution type indicated based on geographic location within a sub-basin unit (Figure 4).

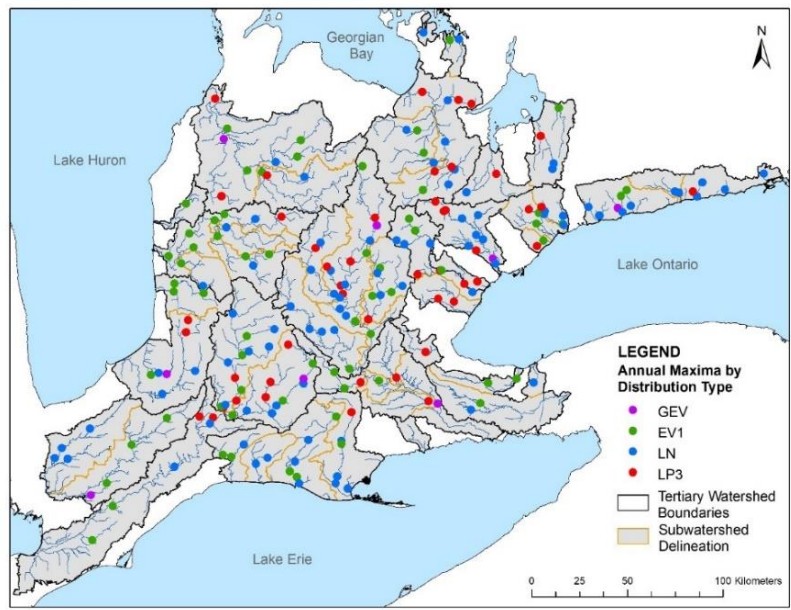

**Figure 4 - Map identifying the geographic location of 207 gauge stations and the optimal statistical distribution selected to model the AMS data. Subdivisions of the tertiary level watershed boundaries are indicated. No sub-basin indicates a single "best fit" distribution type.**

The optimal probability distribution curve is used to estimate the flood quantiles for RIs of 1.25, 2, 5, 10, 25, 50 and 100 years for each of the 207 gauge stations. These flood quantiles are consistent with return periods explored in other flood frequency analyses (Ahn & Palmer, 2016; Basso et al., 2016; Hollis, 1975; Onen & Bagatur, 2017). A Shapiro-Wilk analysis tests the null hypothesis that the flood quantile datasets are normally distributed (Table 1). The dataset for each flood quantile does not meet the assumption of normality ($p<0.05$) and the null hypothesis is rejected. A logarithmic transformation is applied to all flood quantile values. A subsequent Shapiro-Wilk test of the log-transformed flood quantile datasets fails to reject the null hypothesis ($p>0.05$) suggesting the log-transformed flood quantiles are normally distributed.

**Table 1 – Results of Shapiro-Wilks Normality Tests for each flood quantile, with and without logarithmic transformation of data.**

| Test 1 – before transformation | | | Test 2 – after logarithmic transformation | | |
|---|---|---|---|---|---|
| **Flood quantile** | **W-stat** | **p-value** | **Log-transformed flood quantile** | **W-stat** | **p-value** |
| $Q_{1.25}$ | 0.578 | < 2.2e-16 | Log $Q_{1.25}$ | 0.991 | 0.200 |
| $Q_2$ | 0.580 | < 2.2e-16 | Log $Q_2$ | 0.992 | 0.299 |
| $Q_5$ | 0.595 | < 2.2e-16 | Log $Q_5$ | 0.991 | 0.194 |
| $Q_{10}$ | 0.604 | < 2.2e-16 | Log $Q_{10}$ | 0.990 | 0.135 |
| $Q_{25}$ | 0.635 | < 2.2e-16 | Log $Q_{25}$ | 0.991 | 0.260 |
| $Q_{50}$ | 0.634 | < 2.2e-16 | Log $Q_{50}$ | 0.992 | 0.351 |
| $Q_{100}$ | 0.595 | < 2.2e-16 | Log $Q_{100}$ | 0.991 | 0.247 |

## 4.2 Attributes as predictor variables

Twenty-eight (28) attributes are selected (Table 2) to characterize the drainage area conditions representing the topography, precipitation, land use, and hydrological properties from a geomorphological perspective within the sub-basins that influences channel discharge (across all 7 flood frequency quantiles) in terms of the regional geomorphic, hydrologic, topographic and land use properties.

**Table 2 - Twenty-eight (28) variables representing geomorphic, hydrologic, land use, and topographic variability between sub-basins.**

| No. | PREDICTOR VARIABLES | ABBREVIATION | FILE TYPE | SOURCE |
|---|---|---|---|---|
| **Geomorphic Variables: Dominant Surficial Material** | | | | |
| 1 | Glaciolacustrine Clay | %Clay | Raster Dataset | OGS, 2010 |
| 2 | Glacial Till | %Diamicton | Raster Dataset | OGS, 2010 |
| 3 | Glaciofluvial/Glaciolacustrine Gravel | %Gravel | Raster Dataset | OGS, 2010 |
| 4 | Wetland | %Organic | Raster Dataset | OGS, 2010 |
| 5 | Paleozoic or Precambrian Bedrock | %Bedrock | Raster Dataset | OGS, 2010 |
| 6 | Glaciofluvial/Fluvial/Glaciolacustrine Sand | %Sand | Raster Dataset | OGS, 2010 |
| 7 | Glaciolacustrine/Fluvial Silt | %Silt | Raster Dataset | OGS, 2010 |
| **Hydrologic Variables** | | | | |
| 8 | Gauge Drainage Area | logDrainage | Raster Dataset | OMNR, 2005 |
| 9 | Mean Precipitation | Mean_Precip | Point Shapefile | ECCC, 2020 |
| 10 | Precipitation Days | Precip_Days | Point Shapefile | ECCC, 2020 |
| 11 | Mean Rainfall | Mean_Rainfall | Point Shapefile | ECCC, 2020 |
| 12 | Rainfall Days | Rainfall_Days | Point Shapefile | ECCC, 2020 |
| **Land Use Variables** | | | | |
| 13 | Urban Land Use | %Urban | Raster Dataset | OMNRF, 2019 |
| 14 | Tilled Cropland | %Cropland | Raster Dataset | OMNRF, 2019 |
| 15 | Naturalized Land Use | %Naturalized | Raster Dataset | OMNRF, 2019 |
| **Topographic Variables** | | | | |
| 16 | Gradient Mean | Gradient_Mean | Raster Dataset | OMNR, 2005 |
| 17 | Gradient Standard Deviation | Gradient_StDev | Raster Dataset | OMNR, 2005 |
| 18 | Aspect Mode | Aspect_Mode | Raster Dataset | OMNR, 2005 |
| 19 | Stream Length | Stream_Length | Raster Dataset | OMNR, 2005 |
| 20 | Drainage Density | Drainage_Density | Raster Dataset | OMNR, 2005 |
| 21 | Sub-basin Area | WS_Area | Raster Dataset | OMNR, 2005 |
| 22 | Sub-basin Perimeter | WS_Perimeter | Raster Dataset | OMNR, 2005 |
| 23 | Sub-basin Compactness | WS_Compactness | Raster Dataset | OMNR, 2005 |
| 24 | Minimum Elevation | Min_Elevation | Raster Dataset | OMNR, 2005 |
| 25 | Maximum Elevation | Max_Elevation | Raster Dataset | OMNR, 2005 |
| 26 | Elevation Range | Elev_Range | Raster Dataset | OMNR, 2005 |
| 27 | Elevation Mean | Elev_Mean | Raster Dataset | OMNR, 2005 |
| 28 | Elevation Standard Deviation | Elev_StDev | Raster Dataset | OMNR, 2005 |

The upstream drainage area for each georeferenced gauge station is extracted from the hydrologically enforced DEM. A logarithmic transformation is applied to the drainage area variable values to ensure normality ($W=0.994$, $p$-value=0.522). The geomorphic sub-basin attributes are represented by the percentage of the dominant surficial material within the geographic area of each sub-basin: Glaciolacustrine Clay, Glacial Till, Glaciofluvial/Glaciolacustrine Gravel, Wetland, Paleozoic or Precambrian Bedrock, Glaciofluvial/Fluvial/Glaciolacustrine Sand, and Glaciolacustrine/Fluvial Silt (Figure 5a). The hydrological conditions are characterized by an interpolation of Canadian Climate Normals (Figure 5b).

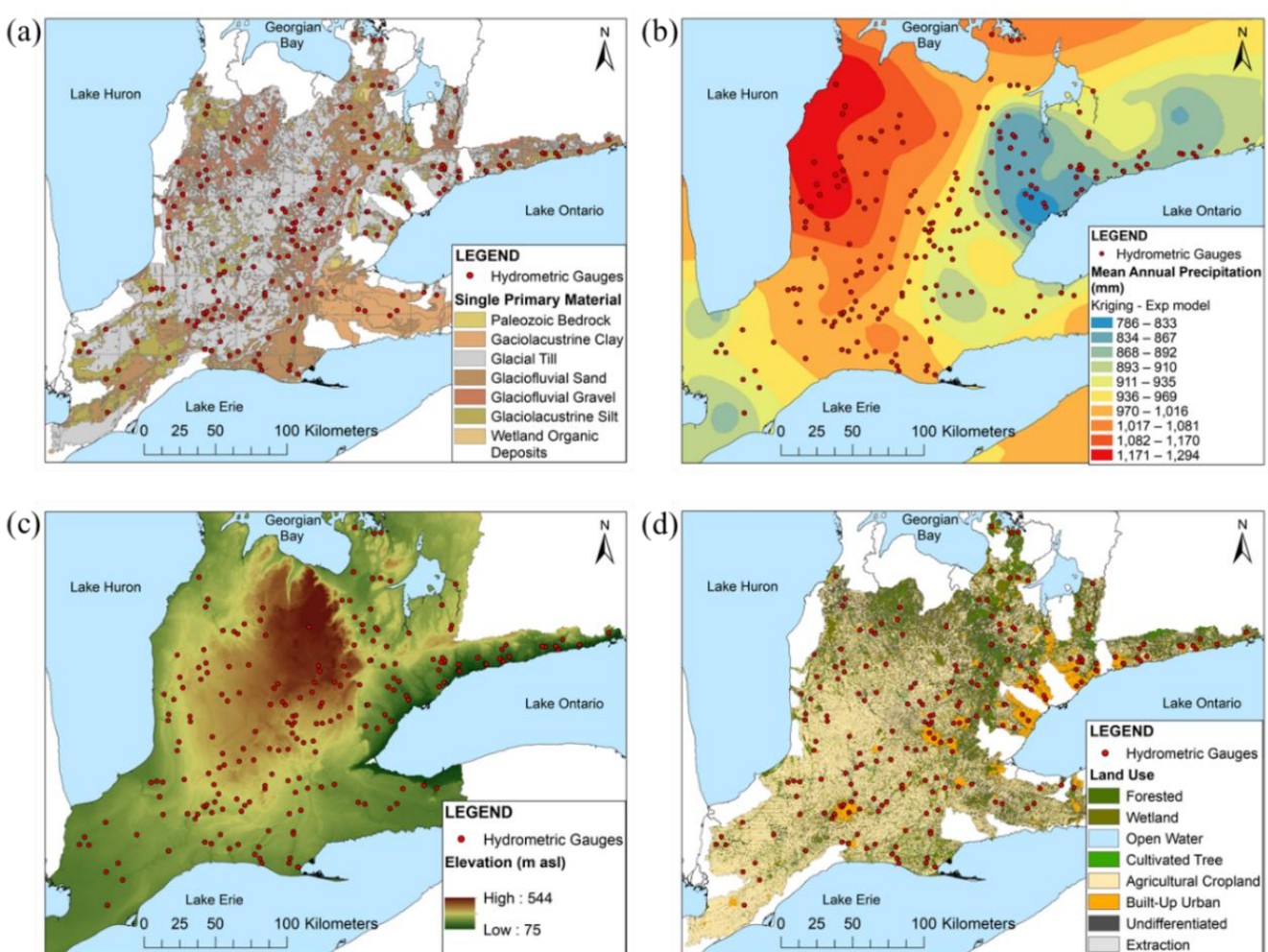

**Figure 5 - Raster datasets illustrating (a) the dominant surficial material, (b) the mean annual precipitation, (c) the hydrologically enforced DEM, and (d) the categorical land use for the peninsular region of southern Ontario.**

Point information for mean annual precipitation, annual number of precipitation days, mean annual rainfall and annual number of rainfall days from 65 observation stations is converted to raster coverage using several interpolation techniques. Inverse distance weighting (IDW) and ordinary kriging (OK) using a stable model and an exponential model are compared. OK has

been shown to produce accurate results when used to describe spatially heterogeneous natural phenomena (Bevan & Conolly, 2009) such as precipitation. Cross validation suggests fitting an OK exponential model for annual mean precipitation, annual mean rainfall, and the annual number of rainfall days, and an OK stable model for the annual number of precipitation days. The topographic conditions of the sub-basins are extracted and quantified from the hydrologically enforced DEM (Figure 5c). Percent land use is quantified using the SOLRIS categories for each sub-basin (Figure 5d). For this study, three land use categories are established: %Urban, %Cropland, and %Naturalized area. %Urban regions combine all transportation and built-up areas. %Cropland is defined by tilled agricultural areas. %Naturalized regions combine all tallgrass landcover, mixed forests, cultivated tree plantations, swamps, wetlands, and open water areas as indicated by the SOLRIS Version 3.0 dataset.

## 5. Results

### 5.1 Single-variate regression RFFA

Regression of the logDrainage variable against each of the seven flood quantile datasets (i.e., $Q_{1.25}$, $Q_2$, $Q_5$, $Q_{10}$, $Q_{25}$, $Q_{50}$, $Q_{100}$) establishes seven single-variable power relationships (Table 3). Statistically significant relations ($p<0.001$) for logDrainage area are indicated across all RIs with a minor, but consistent, decrease in adjusted $R^2$ values as RI increases indicating greater uncertainty in prediction as flood magnitude increases. Research indicates that the $Q_2$ flood quantile (highlighted in Table 3) represents a flow magnitude and frequency that is important to the maintenance of channel morphology and has, therefore, been used in a discharge- drainage area relation in numerous other studies of the southern Ontario region (Annable et al., 2011; Phillips & Desloges, 2014; Thayer et al., 2016; Vocal Ferencevic & Ashmore, 2012).

**Table 3 – Single-variate RFFA models for each flood quantile.**

| Flood Quantile | Single Variable Models (i.e., logDrainage) | | |
|---|---|---|---|
| | Equation | Residual SE | adjusted $R^2$ |
| $Q_{1.25}$ | log$Q_{1.25}$ = −0.858 + 0.945(logDrainage) | 0.217 | 0.867 |
| **$Q_2$** | **log$Q_2$ = −0.746 + 0.957(logDrainage)** | **0.224** | **0.862** |
| $Q_5$ | log$Q_5$ = −0.549 + 0.945(logDrainage) | 0.219 | 0.864 |
| $Q_{10}$ | log$Q_{10}$ = −0.384 + 0.917(logDrainage) | 0.233 | 0.842 |
| $Q_{25}$ | log$Q_{25}$ = −0.223 + 0.906(logDrainage) | 0.245 | 0.824 |
| $Q_{50}$ | log$Q_{50}$ = −0.084 + 0.885(logDrainage) | 0.278 | 0.776 |
| $Q_{100}$ | log$Q_{100}$ = −0.053 + 0.864(logDrainage) | 0.321 | 0.713 |

Expressing the $Q_2$ results (bolded in Table 3) in a power-law format (Eq. (1)), the $Q_2$ model is found to be similar to the findings of other southern Ontario models similarly derived from annual maximum series datasets of the southern Ontario region (Annable, 1995, Phillips & Desloges, 2014). The $Q_2$ power relationship identified in this study indicates a slightly lower estimate of $Q_2$ discharge for smaller drainage areas (<100 km$^2$) compared to the research of others (Figure 6). For larger drainage areas (>100 km$^2$), this study predicts similar discharge estimates compared to the relationship of Phillips & Desloges

(2014) but greater discharge estimates than Annable (1995). Neither of those studies specified best-fit RFFA distributions so the relationship presented here is considered more robust.

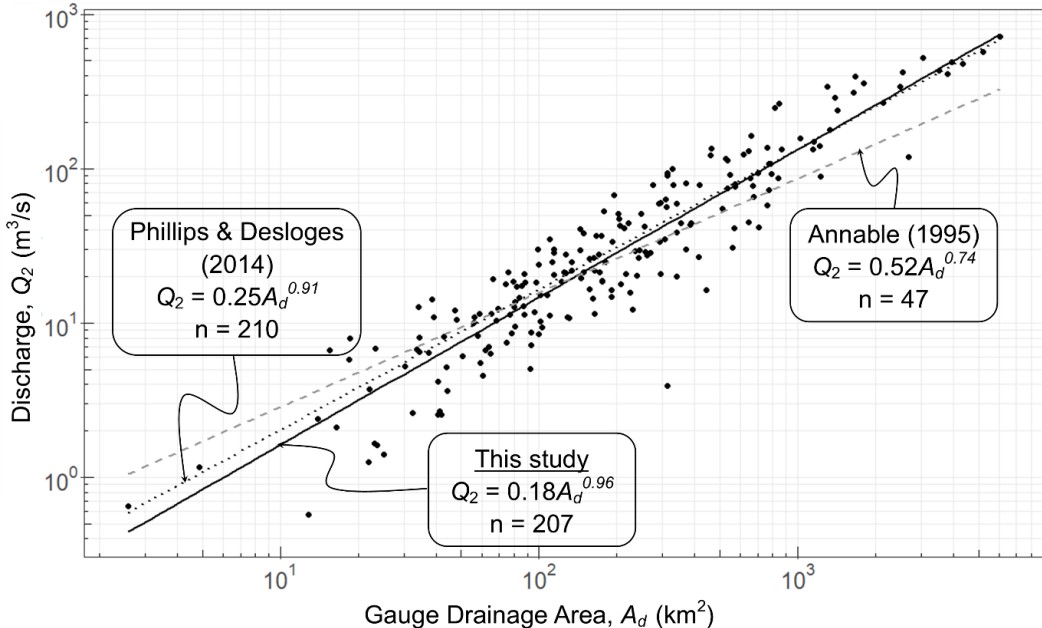

**Figure 6 – The single-variable discharge-drainage area relationship for the $Q_2$ flood quantile of this study (solid black line) compared to the findings of others. The $Q_2$ relationship of Annable (1995) is indicated by the dashed black line. The $Q_2$ relationship of Phillips and Desloges (2014) is given by the dotted black line.**

## 5.2 Multi-variate regression RFFA with parameterized basin characteristics

### 5.2.1. Basin characterization parsimony

A PCA of the 28 explanatory variables produces seven components with eigenvalues greater than 1.00 (eigenvalues for Dim.1 through Dim.7 are 7.85, 5.48, 2.87, 2.25, 1.84, 1.09, and 1.04, respectively) that explain 80.1% of the total variability of the dataset. This suggests an absence of strong intercorrelations among many of the 28 variables. The latent root criterion (also known as the Kaiser or eigenvalue-one criterion) suggests retaining and interpreting principal components if the eigenvalue is greater than 1.00 (Kaiser, 1960). However, using the point where the first few eigenvalues depart from the more similar lesser eigenvalues (i.e., the broken-stick model) (Jackson, 1993), suggests retaining the first 3 dimensions which account for almost 58% of the total variability of the dataset and are the most interpretable. The correlation circles illustrate the projections of the first three principal components (Dim1, Dim2 and Dim3) (Figure 7). Highly correlated variables project in the same direction. The first principal component (Dim1) tends towards a land use composition grouping with some loading from gradient variables and precipitation variables (Dim1 explains 28.0% of the variance). The second principal component (Dim2) tends towards an elevation cluster with additional loading from precipitation variables (Dim2 explains 19.6% of the variance). The third principal component (Dim3) is a weakly defined land surface grouping with loading from surficial geology classifications and basin geometry (Dim3 explains 10.2% of the variance). While elevation is a clear contributor to the variance of the dataset

based on the PCA tests, the directional indicators suggest the presence of multicollinearity among the elevation predictors, reinforcing the results of Pearson correlation detection.

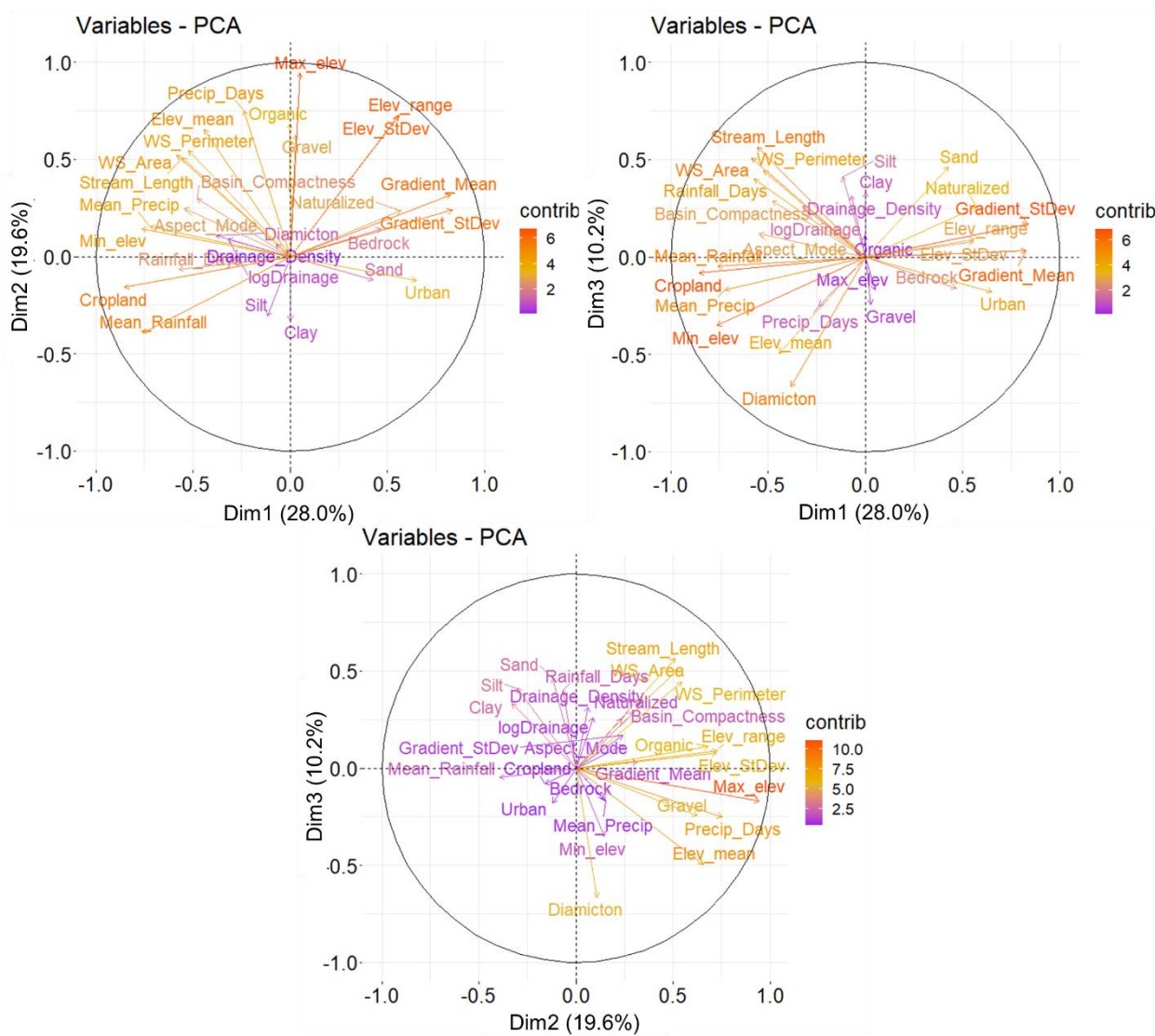

**Figure 7 - Principal component analysis (PCA) highlighting the most contributing variables of the 28-variable dataset for each dimension and illustrating the correlation circles for principal components one, two, and three (Dim1, Dim2 and Dim3).**

A simple Pearson correlation ($|r|>0.6$) suggests that 29 correlated relationships exist among eleven basin characteristics including %Diamicton and %Sand, %Cropland and %Naturalized, Gradient_Mean, Gradient_StDev, Stream_Length, WS_Area, Mean_Rainfall, Min_elev, Max_elev, Elev_mean, and Elev_StDev. PCA and Pearson correlation support elimination of elevation variables (i.e., Min_elev, Max_elev, Elev_mean, Elev_StDev, Elev_Range) except for Gradient_Mean which is retained as the sole predictor to represent the variability of topography among the sub-basins. Correlations between

Gradient_Mean and Gradient_StDev ($r=-0.795$) and between Mean_Rainfall and Mean Precipitation ($r=0.697$) also result in

the elimination of Gradient_StDev and Mean_Rainfall from the potential predictor variables. The Mean_Precipitation variable is retained as it explains snowmelt floods which dominate the flow regime of southern Ontario rivers (Javelle et al., 2003). Other correlated variables are removed from further analysis due to high correlation: the Pearson correlation value, $r$, for WS_Area versus WS_Compactness is 0.717 and the value for Stream_Length versus WS_Perimeter is 0.884. Subyani et al. (2012) similarly find significant correlation between stream length and basin perimeter. The variable WS_Area is removed in favour of WS_Compactness which is a descriptor of basin shape (Apaydin et al., 2006). Although total catchment size has been shown to have a role in the catchment hydrologic response (Merz & Blöschl, 2009), the contributing upstream drainage area is more proportionally relevant to channel discharge at individual gauge stations (Dunne & Leopold, 1978; Prancevic & Kirchner, 2019). Stream_Length is also removed in favour of retaining WS_Perimeter which is a descriptor of basin shape. The retention of WS_Compactness and WS_Perimeter enables greater focus on basin shape which impacts hydrologic relationships and the efficiency with which a fluvial system can evacuate precipitation from the region (Apaydin et al., 2006; Fryirs & Brierley, 2012). For example, elongated catchments (WS_Compactness ~ 0.4) have been shown to have slower runoff (Fryirs & Brierley, 2012). Mean_Rainfall is also eliminated due to correlation with multiple variables (i.e., Mean_Precip, %Cropland, Gradient_StDev). By eliminating nine (9) sub-basin characteristics (Min_Elev, Max_Elev, Elev_Mean, Elev_StDev, Elev_Range, Gradient_StDev, Mean_Rainfall, WS_Area, Stream_Length), 27 of the 29 correlated relationships identified by the Pearson correlation tests (i.e., criterion $|r|>0.6$) are removed. Contrasting geomorphic conditions between catchments (i.e., surficial geology) are represented by a high negative correlation ($r=0.707$) between %Sand or %Diamicton, however, both are initially retained for multi-variate modelling to represent the primary substrate of each sub-basin.

### 5.2.2. Multi-variate regression analysis with backward elimination to identify the most parsimonious models.

It can be a practice to test and transform independent variables to ensure a normal distribution of a multi-variate dataset, however, tests for multi-variate normality are rarely performed (Tacq, 2010). Alternatively, the plots of standardized residuals from combinations of predictor variables are examined for a desired elliptically symmetric distribution (Sheather & Oostrom, 2009. To enable model comparison, the gauge drainage area predictor variable included in the multi-variate regression is logarithmically transformed, consistent with the single-variable power model.

Multiple linear regression is applied to the remaining 19-variable dataset for each of the seven flood RIs, i.e., 1.25, 2, 5, 10, 25, 50 and 100 years, using an OLS approach. The fitted values of the model are compared to the dependent variables (i.e., $Q_{1.25}$, $Q_2$, $Q_5$, $Q_{10}$, $Q_{25}$, $Q_{50}$, and $Q_{100}$) to detect heteroscedasticity. Regression of all 19 predictor variables for each flood quantile reveals statistically significant relationships ($p<2.2e-16$) suggesting at least one of the predictor variables is significantly related to the quantile discharge (Table 4). The multiple coefficients of determination are greater than 0.8 ($R^2>0.8$) for all quantiles.

**Table 4 – Results from model regression (regression of all 19 predictors) indicating predictor variables found to be statistically significant in the 19-variable model. The remaining variables were not found to be statistically significant in 19-variable regressions.**

| 19-Variable Model | Residual S.E. | df | $R^2$ | adj $R^2$ | F-stat | p-value | Variables identified as statistically significant | t-statistic | p-value | Normalized Proportion of Variance |
|---|---|---|---|---|---|---|---|---|---|---|
| $Q_{1.25}$ | 0.152 | 187 | 0.941 | 0.935 | 156.7 | <2.2e-16 | logDrainage | 45.112 | <2.2e-16 | 75.51% |
| | | | | | | | %Naturalized | -1.759 | 0.080 | 2.72% |
| | | | | | | | Gradient_mean | 2.622 | 0.010 | 2.25% |
| $Q_2$ | 0.161 | 187 | 0.935 | 0.929 | 142.5 | <2.2e-16 | logDrainage | 42.947 | <2.2e-16 | 75.35% |
| | | | | | | | Gradient_mean | 2.448 | 0.015 | 2.39% |
| $Q_5$ | 0.1534 | 187 | 0.940 | 0.934 | 153.1 | <2.2e-16 | logDrainage | 44.565 | <2.2e-16 | 75.31% |
| | | | | | | | Gradient_mean | 2.319 | 0.022 | 2.49% |
| $Q_{10}$ | 0.174 | 187 | 0.920 | 0.912 | 113.0 | <2.2e-16 | logDrainage | 38.081 | <2.2e-16 | 74.73% |
| | | | | | | | %Naturalized | -2.074 | 0.039 | 2.73% |
| $Q_{25}$, | 0.186 | 187 | 0.908 | 0.899 | 97.5 | <2.2e-16 | logDrainage | 35.663 | <2.2e-16 | 75.03% |
| | | | | | | | %Organic | -1.999 | 0.047 | 0.67% |
| | | | | | | | %Bedrock | -2.024 | 0.044 | 1.06% |
| | | | | | | | %Naturalized | -1.760 | 0.080 | 2.65% |
| $Q_{50}$ | 0.225 | 187 | 0.868 | 0.854 | 64.5 | <2.2e-16 | logDrainage | 28.982 | <2.2e-16 | 74.51% |
| | | | | | | | %Organic | -2.196 | 0.029 | 0.86% |
| | | | | | | | %Bedrock | -2.227 | 0.024 | 1.44% |
| $Q_{100}$ | 0.273 | 187 | 0.812 | 0.792 | 42.4 | <2.2e-16 | logDrainage | 23.417 | <2.2e-16 | 73.67% |
| | | | | | | | %Organic | -2.228 | 0.024 | 1.09% |
| | | | | | | | %Bedrock | -2.394 | 0.018 | 1.94% |
| | | | | | | | Aspect_Mode | -1.759 | 0.080 | 0.49% |

Examination of the associated *p*-values for the *t*-statistic of each predictor variable indicates that, for all seven flood quantiles, logDrainage overwhelmingly contributes to the 19-variable prediction models (>70%), although its importance decreases as flood frequency decreases. The 19-variable regression (logDrainage included) indicates that either %Naturalized or Gradient_mean is statistically significant for predicting flood quantiles $Q_{1.25}$, $Q_2$, $Q_5$, $Q_{10}$ and $Q_{25}$. This is consistent with the results of the PCA which identifies the first principal component as a land use grouping and the second principal component as an elevation grouping. Other potential predictor variables do not indicate statistical significance in the 19-variable models. During backward elimination, added-variable plots (or partial regression plots) indicating low statistical significance are consistent with the *t*-statistic results confirming a lack of significance. Variables indicating no relationship are eliminated. A high VIF (VIF>5) confirms %Diamicton is unsuitable as a predictor variable due to multicollinearity. Marginal model plots indicate somewhat linear but weak relationships for %Organic, %Sand, and %Gravel. For both percent %Clay and percent %Bedrock, high incidences of zero values produce non-linear relationships suggesting a lack of significance as predictor variables. A high VIF is indicated for %Cropland. High negative correlation (*r*=-0.775) is also observed between %Cropland

and Gradient_Mean resulting in the elimination of %Cropland as a predictor. All five-predictor models demonstrate statistical significance; however, the identified variables are not consistent over all seven flood quantiles (Table 5).

**Table 5 – Regression results of five-predictor models and three-predictor models over all flood quantiles tested. Variables with some explanatory contribution in five- and three-predictor models are indicated by an "X".**

| # of variables | Model | %Gravel | %Organic | %Bedrock | %Sand | logDrainage | Mean_Precip | Precip_Days | Rainfall_Days | %Urban | %Naturalized | Gradient_Mean | Drainage_Densit | WS_Perimeter | Adjusted $R^2$ | Residual SE | $F$-statistic | $p$-value |
|---|---|---|---|---|---|---|---|---|---|---|---|---|---|---|---|---|---|---|
| 5-variable models | $Q_{1.25}$ | | X | | | X | X | | | | X | X | | | 0.924 | 0.164 | 504.6 | <2.2e-16 |
| | $Q_2$ | X | | | X | X | X | | | | X | | | | 0.924 | 0.166 | 502.9 | <2.2e-16 |
| | $Q_5$ | | X | | | X | X | | | | X | | X | | 0.925 | 0.163 | 509.1 | <2.2e-16 |
| | $Q_{10}$ | | | | | X | | X | X | | X | | | X | 0.901 | 0.185 | 374.0 | <2.2e-16 |
| | $Q_{25}$ | | X | | | X | X | | | | X | | | X | 0.886 | 0.198 | 319.9 | <2.2e-16 |
| | $Q_{50}$ | | X | | X | X | | | | | X | | | X | 0.823 | 0.248 | 192.1 | <2.2e-16 |
| | $Q_{100}$ | | X | X | | X | | | | X | X | | | | 0.792 | 0.276 | 153.2 | <2.2e-16 |
| 3-variable models | $Q_{1.25}$ | | | | | X | X | | | | X | | | | 0.922 | 0.166 | 808.9 | <2.2e-16 |
| | $Q_2$ | | | | | X | X | | | | X | | | | 0.919 | 0.171 | 782.5 | <2.2e-16 |
| | $Q_5$ | | | | | X | X | | | | X | | | | 0.923 | 0.165 | 826.8 | <2.2e-16 |
| | $Q_{10}$ | | | | | X | | | X | | X | | | | 0.892 | 0.192 | 567.6 | <2.2e-16 |
| | $Q_{25}$ | | | | | X | X | | | | X | | | | 0.881 | 0.203 | 500.3 | <2.2e-16 |
| | $Q_{50}$ | | X | | | X | | | | | X | | | | 0.821 | 0.249 | 315.9 | <2.2e-16 |
| | $Q_{100}$ | | X | | | X | | | | | X | | | | 0.755 | 0.296 | 212.9 | <2.2e-16 |


The five- and three-predictor models retain surficial material variables (%Organic, %Sand, %Gravel) and climate variables (Mean_Precip, Precip_Days, Rainfall_Days). Reducing from 5-variable models to 3-variable models decreases the adjusted $R^2$ value and increases the $F$-statistic over all flood quantiles. Three-predictor models demonstrate statistical significance; however, the third variable is not consistent over all seven flood frequency quantiles. Results show that including Mean_precip

as a variable increases model fit for flood quantiles $Q_{1.25}$, $Q_2$, $Q_5$, and $Q_{25}$, whereas including Rainfall_Days improves the goodness-of-fit for the $Q_{10}$ model. Alternatively, the inclusion of %Organic is shown to have some statistical significance ($p<0.05$) as a third predictor variable for flood quantiles $Q_{50}$ and $Q_{100}$. For all seven flood quantiles, $Q_{1.25}$, $Q_2$, $Q_5$, $Q_{10}$, $Q_{25}$, $Q_{50}$, and $Q_{100}$, backward elimination reduces to the same single independent variable, logDrainage. As the most parsimonious model, logDrainage is shown to significantly predict discharge ($p<2.2e-16$) confirming the often used single-variable

relationship between drainage area and discharge. However, all seven flood quantiles also identify a two-predictor model using

the variables logDrainage and %Naturalized where *t*-tests demonstrate that both logDrainage and %Naturalized are statistically significant ($p<2.2e-16$) (Table 6).

**Table 6 – The most parsimonious two-variate RFFA models for each flood quantile.**

| Flood Quantile | Two-variable Models (i.e., **logDrainage** and **%Naturalized**) | | |
|---|---|---|---|
| | Equation | Residual SE | adj $R^2$ |
| $Q_{1.25}$ | $\log Q_{1.25} = -0.537 + 0.917(\text{logDrainage}) - 1.179(\%\text{Naturalized})$ | 0.177 | 0.911 |
| $Q_2$ | $\log Q_2 \quad = -0.420 + 0.928(\text{logDrainage}) - 1.196(\%\text{Naturalized})$ | 0.184 | 0.907 |
| $Q_5$ | $\log Q_5 \quad = -0.224 + 0.916(\text{logDrainage}) - 1.194(\%\text{Naturalized})$ | 0.179 | 0.910 |
| $Q_{10}$ | $\log Q_{10} \quad = -0.068 + 0.889(\text{logDrainage}) - 1.158(\%\text{Naturalized})$ | 0.197 | 0.887 |
| $Q_{25}$ | $\log Q_{25} \quad = \quad 0.090 + 0.878(\text{logDrainage}) - 1.148(\%\text{Naturalized})$ | 0.213 | 0.867 |
| $Q_{50}$ | $\log Q_{50} \quad = \quad 0.223 + 0.858(\text{logDrainage}) - 1.127(\%\text{Naturalized})$ | 0.251 | 0.818 |
| $Q_{100}$ | $\log Q_{100} \quad = \quad 0.353 + 0.837(\text{logDrainage}) - 1.100(\%\text{Naturalized})$ | 0.299 | 0.750 |

**5.3 Model evaluation**

For all seven flood quantile models, the addition of the %Naturalized predictor variable reduces the residual standard error (Res SE) and increases the adjusted coefficient of determination (adj $R^2$). Plots of the fitted single-variable models and the two-variable models versus gauge derived estimates of discharge, demonstrate less scatter (Figure 8 illustrates six quantiles). The two-variable model (i.e., logDrainage and %Naturalized) results in an increased adjusted $R^2$ of approximately 0.05 and a

lower standard error suggesting the addition of the second variable (i.e., %Naturalized) improves the goodness-of-fit for all seven flood quantile models (six illustrated). The two-predictor combination of variables provides an improved explanation for the variations in discharge by nearly 5%. Generally, an increase in model scatter is observed for both one-variable and two-variable prediction as the RI increases suggesting increasing uncertainty in prediction of discharge moving from $Q_{1.25}$ to $Q_{100}$. This is consistent with the higher variance in discharge observed for large, infrequent flood events within the original gauge

datasets. The predictability of larger flood events is limited by both the low frequency with which they occur, and the length of the gauge records analyzed (average 42.5 years).

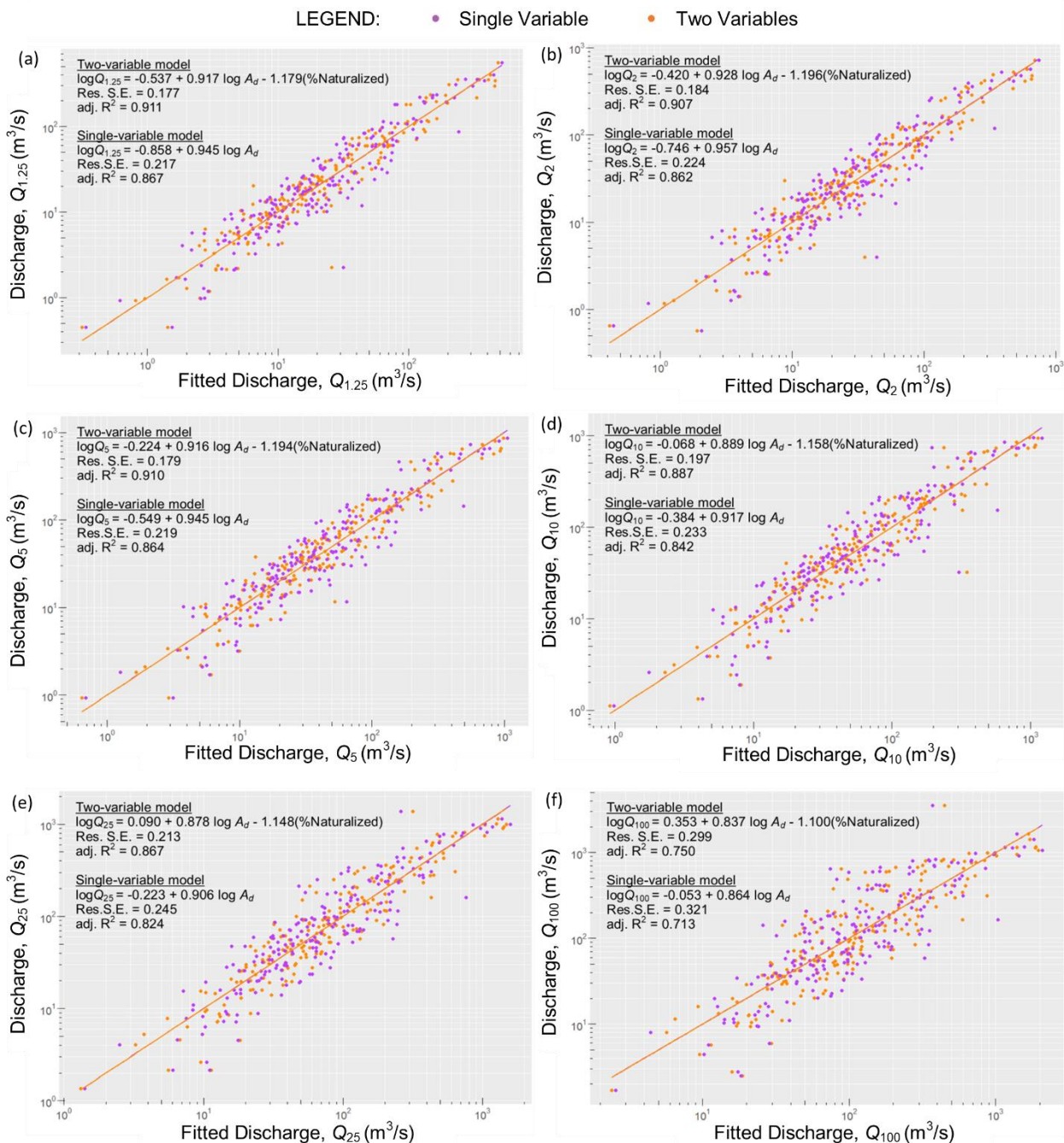

**Figure 8 – The single-variable and two-variable fitted discharge models are plotted against the best-fit estimated discharge derived from the flood frequency curves for (a) a 1.25-year RI, (b) a 2-year RI, (c) a 5-year RI, (d) a 10-year RI, (e) a 25-year RI, and (f) a 100-year RI. The two-predictor logDrainage and %Naturalized model shows less scatter and a lower standard error than the single logDrainage predictor for all six flood RIs shown. The explanation of variability in flood discharge is improved by nearly 5% using the two-predictor model for all seven flood quantiles.**


An analysis of variance (ANOVA) (Table 7) comparing the single-variable models to the two-variable models further indicates an improved prediction of discharge using the two-predictor model compared to the one-predictor model. For all seven flood

quantiles, a decrease in the sum of squares of residuals is observed with the addition of the %Naturalized predictor and an $F$-statistic ($p<0.001$) demonstrates very strong evidence in favour of the two-predictor model. These results are supported by leave-one-out cross validation (LOOCV) (Table 7). For all seven flood quantiles, LOOCV demonstrates a reduction of the root mean square error (RMSE), an improvement of the $R^2$ value, and a reduction of the mean absolute error for the two-variable model compared to the single-variable model.

**Table 7 – Leave-one-out cross validation (LOOCV) and analysis of variance (ANOVA) comparing the single-variable models to the two-variable models.**

| Flood Quantile | Model | Leave-one-out Cross Validation | | | Analysis of Variance (ANOVA) | | | |
|---|---|---|---|---|---|---|---|---|
| | | RMSE | $R^2$ | MAE | RSS | Sum of Sq | F-stat | Pr(>F) |
| $Q_{1.25}$ | Single-variable Model | 0.218 | 0.865 | 0.171 | 9.669 | 3.269 | 104.2 | < 2.2e-16 |
| | Two-variable Model | 0.178 | 0.909 | 0.136 | 6.400 | | | |
| $Q_2$ | Single-variable Model | 0.225 | 0.860 | 0.177 | 10.264 | 3.366 | 99.5 | < 2.2e-16 |
| | Two-variable Model | 0.185 | 0.905 | 0.144 | 6.898 | | | |
| $Q_5$ | Single-variable Model | 0.220 | 0.862 | 0.176 | 9.850 | 3.353 | 105.3 | < 2.2e-16 |
| | Two-variable Model | 0.180 | 0.908 | 0.143 | 6.497 | | | |
| $Q_{10}$ | Single-variable Model | 0.234 | 0.840 | 0.183 | 11.081 | 3.156 | 81.2 | < 2.2e-16 |
| | Two-variable Model | 0.199 | 0.884 | 0.153 | 7.925 | | | |
| $Q_{25}$ | Single-variable Model | 0.247 | 0.821 | 0.193 | 12.343 | 3.102 | 68.5 | 1.65e-14 |
| | Two-variable Model | 0.214 | 0.865 | 0.169 | 9.241 | | | |
| $Q_{50}$ | Single-variable Model | 0.279 | 0.773 | 0.215 | 15.859 | 2.987 | 47.3 | 7.18e-11 |
| | Two-variable Model | 0.253 | 0.814 | 0.198 | 12.872 | | | |
| $Q_{100}$ | Single-variable Model | 0.322 | 0.709 | 0.248 | 21.127 | 2.848 | 31.8 | 5.70e-08 |
| | Two-variable Model | 0.301 | 0.746 | 0.237 | 18.279 | | | |

## 6. Discussion

Flood magnitude, frequency and duration are primary drivers of channel erosion and stream morphology (Taniguchi & Biggs,

2015). High-magnitude, less-frequent floods will undoubtedly result in significant alterations to a channel's morphology and are more important when considering hazards, loss of life and infrastructure damage (Onen & Bagatur, 2017), however, the cumulative effects of more frequent, lower-magnitude floods can also be geomorphically more effective in altering channel form (Church & Ferguson, 2015; Wolman & Gerson, 1978; Wolman & Miller, 1960). Consequently, for effective risk

management and hazard prevention, it is useful to model flows of different flood RIs when considering flood frequency as a
predictive tool to better understand a river's morphological response to discharge (Basso et al., 2016*).* The best estimation of extreme flood events, however, is limited by the availability and accuracy of recorded gauge data, the length of the observed flood series, and the presence or absence of extreme flood occurrences within a flow record (Odry & Arnaud, 2017). This analysis uses a broad range of high- and low-frequency flood estimates from long-term historical flow data to develop a reliable RFFA for urban planning and infrastructure engineering. It is common practice to develop an RFFA relating the drainage area
of a catchment to channel discharge using a single-variable power-law relationship. Research suggests that physiographic features, such as those inherited by southern Ontario's glacial legacy, and anthropogenic land use, for example southern Ontario's clustered urbanization and widespread agricultural development, can influence a region's hydrogeomorphic response, particularly in smaller watersheds (Royall, 2013). Seeking to improve upon a widely accepted single-variate RFFA model in a heterogenous landscape, the objective of this study was to explore a dependable RFFA using a multi-variate
approach for a region influenced by glacial conditioning and varying land use, while also considering the hydrologic influences of climate and topography.

In this study, rigorous goodness-of-fit testing of annual maximum mean daily discharge data series from 207 hydrometric gauge stations in a heterogeneous landscape shows that 42.5% of gauge records are most suited to a 2-parameter LN distribution, 31.9% to a 2-parameter EV1 distribution, 21.7% to a 3-parameter LP3 distribution, and 3.9% to a 3-parameter
GEV distribution. This suggests that all four distributions are potentially suitable for modelling flood extremes in heterogenous regions. The model selection criteria favoured a 2-parameter model over a 3-parameter model in 74.4% of cases, consistent with other studies which found that selection criteria demonstrate a predisposition towards the most parsimonious model (i.e., fewest distribution parameters) (Farooq et al., 2018; Laio et al., 2009; Onen & Bagatur, 2017). Most notably, the 2-parameter EV1 model is optimal five (5) times more frequently than its 3-parameter parent model, the GEV distribution, which is only
found appropriate for use in 3.9% of cases. This finding is similar to that of Laio et al. (2009) where the GEV distribution was only selected in a limited number of cases when modelling the annual maxima of peak discharge in 1000 United Kingdom basins. However, the GEV and LP3 distributions are heavier tailed than the LN or EV1 distributions (El Adlouni et al., 2008; Merz et al., 2022; Papalexiou et al., 2013) suggesting the upper tail behaviour of a flood time series may be underestimated when estimating flood frequency from small sample sizes (less than 50) while a single extreme flood event may lead to
overestimation of the upper tail (Papalexiou et al., 2013). The average hydrologic record in this study is 42.5 years which implies uncertainty in estimating extreme quantiles in the study region due to limited record length of some gauges, but research has indicated that basins with snowmelt-dominated regimes tend toward lighter tailed distributions (Merz et al., 2022).

Flood estimation will often apply a universal, fixed probabilistic model to historical gauge data (Di Baldassarre et al., 2009). Other southern Ontario studies have employed a blanket LP3 probability distribution to model the $Q_2$ flood frequency
(Annable, 1995; Phillips & Desloges, 2014). However, the variation of statistical distributions identified as an optimal fit in this study suggests a need for careful, systematic model selection criteria when fitting observed flow data in regions with

variable land use or other hydraulic influences (i.e., geomorphology, substrate materials, climate, or topography). To prevent an over-estimation or, more importantly, an under-estimation of discharge when predicting flood recurrence, model goodness-of-fit should be evaluated. The results of this study indicate that a 2-parameter LN statistical distribution will provide an optimal fit for 43% of the southern Ontario flood records when a broad range of flood quantiles are being examined.

Other studies have explored a variety of novel regionalization approaches. Di Lazzaro et al. (2015) presented an RFFA using a single-variable parameterization of drainage density. Ahn and Palmer (2016) estimate flood frequency using the GEV distribution and then proposed regionalization methods using a spatial proximity approach. However, regionalization based on spatial proximity assumes that nearby sites are more similar than distal sites (Odry & Arnaud, 2017). In a glacially conditioned landscape, such as the southern Ontario region, the configuration of glacial deposits (Figure 5a) often forms drainage divides that segregate neighbouring catchments with diverse flood characteristics. This study, therefore, explores regionalization through a multi-variate regression-based approach to capture the variability of upstream hydrologic controls that are often dependent on the spatial arrangement of post-glacial physiographic features and, in the case of southern Ontario, the variable land use (i.e., regionally clustered urbanization and agricultural development). The mapping of surficial material, climate conditions, topography, and land use illustrates the variability of hydrologic influences on the region (Figure 5). Consistent with the agricultural land use of southern Ontario, analysis reveals a negative correlation between %Cropland and Gradient_mean. Regions of steep gradient are not typically associated with areas of high agricultural activity, whereas lower gradient regions provide much of the agricultural/cropping activity. Crops are typically cultivated in areas with favourable conditions for growth (i.e., rainfall and gradient) producing collinear relationships with key elevation and precipitation variables relevant to channel discharge. Likewise, the high spatial variability in surficial geology of southern Ontario (due to its glacial conditioning) can be problematic. Contrasting geomorphic conditions between catchments are represented by, for example, high negative correlations among %Diamicton and %Sand and an absence of surficial material types in many areas (e.g., %Bedrock and %Clay) produces high incidences of zero values and non-linear relationships. Conversely, a measure of natural land use is available across the study region, making a linear relationship between %Naturalized and discharge possible.

During the backward elimination process, different land use, geomorphic, climatic, and topographic variables assume different importance in predicting channel flow depending on the flood magnitude being modelled. The influence of the glacial legacy is captured by the inclusion of surficial materials in the five- and three-variable models (Table 5). The less parsimonious, but still statistically valid, five- and three-predictor models show the importance of land cover/glacial legacy (%Organics, %Sand, %Gravel) and climate variables (rainfall days). In three-variable models, precipitation (i.e., Mean_precip or Rainfall_Days) increases model fit for lower magnitude, more frequent flood events (i.e., $Q_{1.25}$, $Q_2$, $Q_5$, $Q_{10}$, and $Q_{25}$), suggesting a greater predictive relationship of channel discharge. In contrast, surficial geology (i.e., %Organic) has more predictive value for high magnitude, less frequent flood events ($Q_{50}$ and $Q_{100}$). During low-magnitude flood events, it is unsurprising that a fluvial system's hydrological response is more directly related to the amount of rainfall or snowmelt infiltration whereas during less frequent, high-magnitude or flash flood events, surface saturation across an increasing area of the watershed is more closely

tied to surficial material properties that limit or enhance infiltration, impacting surface runoff. The surficial material %Organic is retained for high-magnitude, low frequency floods (i.e., $Q_{50}$ and $Q_{100}$) in the 3-variable models suggesting that the percentage of a basin with highly organic surficial material (e.g., wetlands) can effectively increase infiltration, limiting overland flow and the magnitude of channel discharge during high magnitude flood events. Surficial material with higher organic content has been shown to significantly increase infiltration capacity and porosity (Luna et al., 2018).

Although the most parsimonious model for estimating discharge is found to be the generally accepted and efficient single-variable relation between discharge and drainage area, when considering model variance, the two-predictor combination of upstream drainage area and the regional percentage of naturalized landscape (%Naturalized) shows a 5% improvement when explaining variation in flood discharge for all RIs tested (i.e., 1.25, 2, 5, 10, 25, 50, and 100 years). An analysis of variance (ANOVA) further indicates a statistically significant improvement in prediction of discharge using the two-predictor model

(i.e., logDrainage and %Naturalized) compared to the single-predictor model (i.e., logDrainage). The percentage of naturalized landscape is important because it reflects areas within a catchment that have enhanced water storage compared to urban or agricultural areas. These findings are important for situations when it is necessary to reduce uncertainty in flood prediction. Plots comparing the single- and two-predictor models demonstrate less scatter for all seven flood quantiles. Generally, an increase in model scatter is observed for both one-variable and two-variable prediction as the RI increases suggesting the

predictive capability lessens moving from $Q_{1.25}$ to $Q_{100}$. This finding is similar to that of Basso et al. (2016) where model performance is better for short and intermediate return intervals. Any flood frequency analysis is limited by the length of the flow records being analysed. Since the average length of gauge records used in this study is 42.5 years, a decrease in model reliability is anticipated as the non-linear hydrological processes of the region are extrapolated. Despite careful selection of the candidate statistical distributions to "best fit" the observed flow records, the absence of large flood events captured within

the sample data can skew the estimation of flood frequency for low-probability, low-frequency events (Odry & Arnaud, 2017).

     The findings of this research demonstrate that land use has greater predictive power than surficial geology when coupled with drainage area to estimate channel discharge in a heterogeneous landscape over a broad range of flood quantiles. While the methodology used in this study is transferable to other regions, this finding may also be transferable. However, a new scenario would require recalibration of the drainage area relationship and possibly reclassification of land use types to suit the spatial

variation of the new location. Human landscape alterations that impact drainage density will influence rates of overland flow and channel flow, exerting additional influence on hydrological processes and stream response and, subsequently, impacting the magnitude and frequency of peak channel flows (Taniguchi & Biggs, 2015). Changes to land cover, such as deforestation, conversion to cropping and urbanization, typically decrease infiltration which increases discharge, and alters flood magnitude (Chin et al., 2013; Royall, 2013). It follows that the presence of reforested or natural areas will have a significant influence on

modelled discharge. Since the early 1900s, select areas of southern Ontario have been reforested in recognition of wasteful clearing of marginal and submarginal agricultural lands by early settlers (Armson et al., 2001). The %Naturalized variable includes tallgrass landcover, mixed forests, cultivated tree plantations, swamps, wetlands, and open water areas, representing

areas of high infiltration or the surface storage of water. The negative coefficient for the percentage of naturalized area reduces the weight of the drainage area input. This is consistent with the theoretical expectation that drainage area of sub-basins with

a high percentage of naturalized areas may be overemphasized without the appropriate correction for surface water storage. Although urbanization has been shown to have the most profound influence on fluvial system response, altering hydrological processes through a decrease in infiltration, an increase in overland flow and a potential decrease in groundwater recharge (Chin et al., 2013), the regional impact of clustered urban populations of southern Ontario is diluted by the expansive regions of cropland, grazing, and naturalized areas that separate them. Consequently, the percent %Urban variable shows minimal

significance in the multivariate regression. Similarly, the %Cropland was shown to be a poor regional predictor for discharge due to a collinear relationship with other predictors. The statistical significance of %Naturalized, however, suggests that the percentage of a sub-basin that is naturalized can be an effective variable to represent temporary surface water storage, limiting the impact to a channel during flood events.

## 7. Conclusion

To transfer flood discharge information from gauged sites to ungauged sites in a heterogeneous landscape (e.g., a low-relief, glacially conditioned landscape with variable land use), the primary objective of this research is to explore additional explanatory hydrologic controls to improve the predictive strength of a well-known regional flood frequency approach that correlates drainage area to discharge. The main conclusions of this analysis are:

1) When modelling the annual maximum mean daily discharge records for southern Ontario, 42.5% were most suited to
a 2-parameter LN distribution, 31.9% to EV1, and 21.7% to LP3, and 3.9% to a GEV distribution suggesting all four distributions tested are potentially suitable for modelling flood extremes in a heterogeneous landscape. The variation of "best fit" probability distributions indicates that systematic model selection criteria is necessary when fitting observed flow data in regions with variable land use or other hydraulic influences (i.e., geomorphology, climate, or topography).

2) For lower magnitude, more frequent flood events (i.e., $Q_{1.25}$, $Q_2$, $Q_5$, $Q_{10}$, and $Q_{25}$), precipitation shows a greater predictive relationship with channel discharge in 3- and 5-variable models whereas for high magnitude, less frequent flood events surficial geology has more predictive value. For high-magnitude, low frequency floods (i.e., $Q_{50}$ and $Q_{100}$) highly organic surficial material (e.g., wetlands) can effectively increase infiltration, limiting overland flow and the magnitude of channel discharge during high magnitude flood events.

3) While land use, geomorphology, material type, climate, and topographic variables are variably important on the flood magnitude being modelled, the results here show the most parsimonious predictor for estimating discharge in ungauged streams is the accepted and efficient single-variable, drainage area.

4) However, when considering model variance, a two-predictor combination of upstream drainage area and the regional percentage of naturalized landscape shows a statistically significant 5% improvement when explaining variation in

flood discharge for a broad range of recurrence intervals tested (i.e., 1.25, 2, 5, 10, 25, 50, and 100 years). The negative coefficient associated with the percentage of naturalized area reduces serves as a correction to the drainage area relationship to account for surface water storage. This finding is important for situations when it is necessary to reduce uncertainty in flood prediction.

In summary, the findings suggest that applying a zonal two-variable model, which accounts for drainage area and the
percentage of upstream naturalized land use, serves as a correction for surface water storage when modelling flood magnitude for high- and low-frequency flood events. This improvement is of value in a heterogeneous landscape when considering the geomorphic response of channels to predicted channel discharge for a broad range of flood recurrence intervals and greater precision is required.

## Appendix A: Probability distribution functions

The GEV distribution uses a three-parameter probability distribution function such that

$$F(x) = \begin{cases} exp\left(-\left(1 - \varepsilon\frac{x-\mu}{\sigma}\right)^{1/\varepsilon}\right) & \varepsilon \neq 0 \\ exp\left(-exp\left(-\frac{x-\mu}{\sigma}\right)\right) & \varepsilon = 0 \end{cases} \tag{A.1}$$

where $\mu$, $\sigma$ and $\varepsilon$ are the location, scale, and shape parameters of the flow data, respectively. The location parameter describes the shift of a distribution along the horizontal axis, while the scale and shape parameters describe the spread (Zhang et al., 2020). The GEV blends the Gumbel (EV1), Frechet and Weibull distributions which are nested models within the GEV
distribution (Laio et al., 2009). The simplified EV1 distribution uses the GEV function where the shape parameter, $\varepsilon$, is reduced to zero, giving the two-parameter probability distribution function

$$F(x) = exp\left(-exp\left(-\frac{x-\mu}{\sigma}\right)\right) \tag{A.2}$$

where $\mu$ is the location parameter and $\sigma$ is the scale parameter. Consideration of the three-parameter GEV distribution balances model bias versus model variance. The more complicated three-parameter GEV distribution reduces model bias compared to
the two-parameter EV1 distribution, however, as the number of parameters increases, variance typically increases (Laio et al., 2009). The LN distribution is the log-transformed two-parameter Normal or Gaussian distribution represented by the probability distribution function

$$F(x) = \frac{1}{\sigma\sqrt{2\pi}} exp\left(-\frac{1}{2}\left(\frac{x-\mu}{\sigma}\right)^2\right) \tag{A.3}$$

also applying $\mu$ and $\sigma$ as location and scale parameters, respectively. Similarly, the LP3 distribution is the log-transformed
three-parameter Gamma or Pearson Type III identified by the probability distribution function

$$F(x) = \frac{1}{|\sigma|\Gamma\varepsilon}\left(\frac{x-\mu}{\sigma}\right)^{\varepsilon-1} exp\left(-\frac{x-\mu}{\sigma}\right) \tag{A.4}$$

where $\mu$, $\sigma$ and $\varepsilon$ are the location, scale, and shape parameters, respectively. Pearson Type III and Normal distributions are converted to LP3 and LN distributions when the data are log-transformed at the outset (Di Baldassarre et al., 2009).

**Data availability statement:**
The data that support the findings of this study are available from the corresponding author, PET, upon reasonable request.

**Author contributions:**
**Pamela E. Tetford**: Conceptualization (lead); methodology (lead); investigation (lead); formal analysis (lead); writing – original draft (lead); writing – review and editing (equal).
**Joseph R. Desloges:** Conceptualization (supporting); methodology (supervision); investigation (supervision); formal analysis (supervision); writing – original draft (supporting); writing – review and editing (equal).

**Competing interests:**
The authors declare that they have no conflict of interest.

**Acknowledgements:**
This work was supported by the Natural Sciences and Engineering Research Council of Canada (NSER-CGS-D graduate scholarship to P. E. Tetford), and research funding from the University of Toronto to J. R. Desloges. Many thanks to Esther Bushuev who helped compile data of the WSC flow gauges for analysis. Academic advice and insights are greatly appreciated from Marney Isaac, Carl Mitchell, and Michael Widener. We also thank our anonymous peer reviewers for their insightful comments that improved the manuscript.

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
