# Peer review of "Modelling flood frequency and magnitude in a glacially conditioned, heterogeneous landscape: testing the importance of land cover and land use."

_Hydrology and Earth System Sciences, 2022_

## Referee Comment (RC1)

The authors present a study that implements an additional environmental variable into a commonly used approach for RFFA through a series of statistical methods. The objective is to incorporate the spatial heterogeneity of basins into the approach, which is interesting in its nature. The results demonstrate improvement, and various variables are utilized in statistical methods to produce more rational results. I have two major comments regarding the description that I believe the authors could address. Furthermore, I have listed several minor comments in order to maximize the impact of this study.

Major comments:
1. The methods section of the current manuscript is somewhat scattered, with several critical details not being explicitly stated and only being supplemented in the results section. As a result, the reader may find it challenging to understand the entire procedures used in this study and would require a considerable amount of effort to piece together all the information, while some crucial details may still be missing. I have listed some examples below, but there may be more. Addressing these concerns would aid in maximizing the reproducibility of this method.
   - L210: It is not clear how the authors define "the theoretical association." I believe it refers to the Pearson correlation, which is mentioned much later in the paper.
   - Section 3.4 consists of only one sentence, which indicates that several details are missing.
   - L218-220: It is unclear how the spatial patterns of 45 sub-basins were incorporated into the analysis of 207 gauges. Additionally, it is not clear whether "sub-watersheds" and "sub-basin units" refer to the same concept or different concepts.
   - L221-222: This information should have been included in Section 3.1 because it pertains to the fundamental information regarding the hydrological data used in the study.
   - L224-226: This information should also have been included in Section 3.1 because it pertains to a strategy used in the study for applying the model selection criteria. The first part of this information is already included in Section 3.1, while the second part is in the results section, leading to confusion. Additionally, the authors should provide relevant references.
   - L235: This sentence is unclear and difficult to comprehend.
   - L258: It is unclear how the authors define "strongly significant."
   - Section 4.3.1 is a method, rather than results.
2. The authors aimed to improve the RFFA method by incorporating variables that consider the spatial heterogeneity of sub-basins and investigated the influence of 28 variables on the RFFA. While the final results indicated an overall 5% improvement in adjusted R2, with the value increasing from approximately 0.85 to 0.9, the authors employed several

statistical methods to eliminate 27 out of the 28 variables. The backward elimination process was used to identify the most critical variables and remove those that did not significantly contribute to the model. However, the authors' justification for discarding the 27 variables is not entirely convincing, especially considering that 27 is a significant portion of the total 28 variables investigated (e.g., the relatively confusing statement in lines 347-370). To reinforce the validity of this analysis, a clear and systematic procedure should be presented (as discarding such a large portion of variables without sufficient justification may raise questions about the validity and robustness of the proposed method), which currently appears to be in a relatively random order. The inclusion of tables or hierarchical diagrams could potentially aid in clarifying the methodology. Additionally, the authors should explain how they selected the different but parallel statistical methods, such as correlation analysis and PCA, and how the elimination process was objectively determined. By reorganizing the description in a more structured and coherent way, the authors could strengthen the justification for the proposed model.

Minor comments:
1.  It is not clear by which method the authors used for comparing the 3(or 4 in some cases) model selection criteria (i.e., AIC, BIC, ADC, AICc) for determining the best fit distributions.
2.  Concerning the Pearson correlation: (1) why use a standard of 0.6 (instead of a common one, e.g., 0.5)? any references? (2) Does the normality been checked? If not, would using the Spearman correlation be a better option
3.  L327-331 stated that WS_Area and Stream_Length are removed whereas WS_Perimeter and Basin_Compactness are retain due to the intension to focus on basin shape rather than basin size for concentrating on the efficiency of the fluvial system transport. However, it should be aware that the length-area relationship has been found as a function of basin shape (Sassolas-Serrayet et al., 2018). Additionally, several studies have shown that basin size plays a critical role in catchment hydrological response and determines the curve of the flood frequency analysis (Merz & Blöschl, 2009). While it is recognized that the drainage size appears in the final model based on its origin as a one-variable model, the description provided by the authors may be misleading.
4.  It is unclear why drainage size still appears in the 28 variables if it has already been recognized as a prominent variable in previous RFFA.
5.  The link between the interpretation of the PCA results and Figure 6 needs improvement, and it is unclear what procedure/standard the authors used to objectively eliminate variables based on the results of the Pearson correlations and PCA
6.  "the observed correlation" in L326 should be stated in a quantitative way.
7.  It is stated in L227-233 and L428-436 that the results of the best-fit distributions show an extreme low percentage for GEV distributions and the authors refer (or hint?) this to the

number of parameters. Although it is not incorrect, it should be noted that the two two-parameter distributions used in this study (EV1 and LN) have lighter-tailed behavior than the two three-parameter distributions (GEV and LP3) (El Adlouni et al., 2008). Based on their results, it implies that the estimates of the extreme flood quantiles are relatively small in this region. Such a result may be due to the shorter dataset used in this study, which should be roughly between 37-48 years long based on the informative mean and standard deviation of the record length provided in this paper. As a result, there may be considerable uncertainty in estimating extreme quantiles for GEV (e.g., Papalexiou and Koutsoyiannis, 2013), which should be properly referenced in the discussion.

8. The title suggests a potential linkage or interaction between these conditions and flood frequency analysis, so it is expected that more discussion of this topic would be included (which is still weak, e.g., 450-452, and supporting references are expected). Otherwise, it may be advisable to reconsider the title to avoid any potential for confusion.

9. In L463-464, the authors imply that they intend to implement another modified model to properly capture spatial variability in a glacial region, but the resulting model obtained in this study eventually excluded all these variables. Please comment on this, and it is not clear whether the scope of this study is "glacial conditions" or "land use" or both from the current title.

10. Please provide more information on the dataset used in Sec. 3.3, such as whether the same basin boundary was used for the analyses of the one-variable model and the two-variable model, the duration of streamflow data, and whether the analysis is representative if the duration of each variable does not overlap (as shown, the precipitation data is from 1981-2000, whereas the land use data is based on the dataset from 2014-2017).

11. L160-161: Since the authors explain the reason for using annual maxima instead of partial duration series (PDS), it would be appropriate to acknowledge other commonly used metrics, e.g., the peak-over-threshold (POT) analysis.

12. There is no clear discussion in the previous section concerning the statement in L537-538, which appears as the closing sentence of the conclusion.

13. It could be beneficial for the authors to consider providing some additional discussion on how the findings of this study may be transferable to other regions. Do the authors suggest directly including naturalized variables in RFFA for other regions, or do they recommend conducting an entire eliminating process based on different environmental variables obtained in the study area?

14. Throughout:
    – There appear to be a relatively large number of minor mistakes, such as unclear meanings of italics and special font sizes (e.g., L83, L86, L93, L310-312, L365, L366), using abbreviations before their definitions (e.g., L16, L87), mistakes in figures (e.g.,

the legend of Fig. 2: HYDAT is not a GIS dataset; the y-ticks of Fig. 7 are missing; Dim1 explains 27.7% in L318 but 28% in Fig. 6), and typos and grammatical errors. It may be beneficial to have someone do a thorough proofreading to address these issues.

- It is suggested that several statements should provide supporting references (e.g., L155, L225 (n/p<40), L469-471).

Reference

El Adlouni, S., Bobée, B., and Ouarda, T. B. M. J.: On the tails of extreme event distributions in hydrology, J. Hydrol., 355, 16–33, https://doi.org/10.1016/j.jhydrol.2008.02.011, 2008.

Merz, R. and Blöschl, G.: Process controls on the statistical flood moments - a data based analysis, Hydrol. Process., 23, 675–696, https://doi.org/10.1002/hyp, 2009.

Papalexiou, S. M., Koutsoyiannis, D., and Makropoulos, C.: How extreme is extreme? An assessment of daily rainfall distribution tails, Hydrol. Earth Syst. Sci., 17, 851–862, https://doi.org/10.5194/hess-17-851-2013, 2013.

Sassolas-Serrayet, T., Cattin, R., and Ferry, M.: The shape of watersheds, Nat. Commun., 9, 1–8, https://doi.org/10.1038/s41467-018-06210-4, 2018.

---

## Author Response (AR1)

We thank the reviewers for the insights they have provided regarding our research paper, "Modelling flood frequency and magnitude in a glacially conditioned, heterogeneous landscape: land use matters".

References to line numbers below are to the revised manuscript.

**Reviewer #1**

Major comments:

1.  *"The methods section of the current manuscript is somewhat scattered, with several critical details not being explicitly stated and only being supplemented in the results section…"*

    We agree with Reviewer #1 that some details needed to be more explicitly stated and we have made the following editorial changes:

    *"L210: It is not clear how the authors define "the theoretical association." I believe it refers to the Pearson correlation, which is mentioned much later in the paper."*

    We have revised the Methods section to explicitly discuss the strategy for applying Pearson correlation (lines 231-233).

    *"Section 3.4 consists of only one sentence, which indicates that several details are missing."*

    We have consolidated Section 3.4 with a new Section 3.3 "Comparison of single-variable RFFA to multi-variate RFFA" that outlines the strategies for testing predictor variable independence, backward elimination, and model evaluation which include an analysis of variance (ANOVA) and the new leave-one-out cross validation (LOOCV) (lines 236-238).

    *"L218-220: It is unclear how the spatial patterns of 45 sub-basins were incorporated into the analysis of 207 gauges. Additionally, it is not clear whether "sub-watersheds" and "sub-basin units" refer to the same concept or different concepts."*

    For clarity, all references to "sub-watersheds" have been replaced with "sub-basins". A revised description of the subdivision of catchments and characterization of sub-basin attributes appears in Methods, Section 3.2.2 (lines 195-200).

    *"L221-222: This information should have been included in Section 3.1 because it pertains to the fundamental information regarding the hydrological data used in the study."*

    We agree with Reviewer #1 that this fundamental information should be included in the Methods section and have included it in Section 3.1 (lines 159-160).

    *"L224-226: This information should also have been included in Section 3.1 because it pertains to a strategy used in the study for applying the model selection criteria. The first part of this information is already included in Section 3.1, while the second part is in the results section, leading to confusion. Additionally, the authors should provide relevant references."*

    We agree with Reviewer #1 that this information should also be included in the Methods section and have included it in Section 3.1 (lines 160-162).

    *"L235: This sentence is unclear and difficult to comprehend."*

    The caption has been edited for clarity (line 253).

    *"L258: It is unclear how the authors define "strongly significant."*

"Strongly significant" has been quantified to demonstrate the statistically significant relationships (p<0.001) between log-drainage area and log-discharge for all return intervals tested (lines 296-298).

*"Section 4.3.1 is a method, rather than results."*

We agree with Reviewer #1 that this information should not be included in the Results section. We have presented a separate section for data input, Section 4 (as per a recommendation from Reviewer #2) and included this information as variable input in Section 4.2 (lines 273-292).

2.  *"The authors aimed to improve the RFFA method by incorporating variables that consider the spatial heterogeneity of sub-basins and investigated the influence of 28 variables on the RFFA…To reinforce the validity of this analysis, a clear and systematic procedure should be presented (as discarding such a large portion of variables without sufficient justification may raise questions about the validity and robustness of the proposed method), which currently appears to be in a relatively random order. The inclusion of tables or hierarchical diagrams could potentially aid in clarifying the methodology. Additionally, the authors should explain how they selected the different but parallel statistical methods, such as correlation analysis and PCA, and how the elimination process was objectively determined. By reorganizing the description in a more structured and coherent way, the authors could strengthen the justification for the proposed model."*

We agree with Reviewer #1 that a more structured presentation of the backward elimination strategy strengthens the justification for the proposed models. We have provided steps for the backward elimination strategy that includes testing for independence, multi-variate diagnostics, and model evaluation (including LOOCV), and we have included a flowchart (Figure 3) illustrating the iterative process for backward elimination of variables (lines 225-241).

Minor comments:

*"It is not clear by which method the authors used for comparing the 3(or 4 in some cases) model selection criteria (i.e., AIC, BIC, ADC, AICc) for determining the best fit distributions."*

Three model selection criteria are used to determine "best fit" for each gauge dataset (i.e., ADC, BIC, and AIC or AICc). For clarity, we have included the following statement regarding model selection in the methods section: "The MSClaio2008 function in R returns the distribution that is most often selected by the model criteria" (lines 179-180).

*"Concerning the Pearson correlation: (1) why use a standard of 0.6 (instead of a common one, e.g., 0.5)? any references? (2) Does the normality been checked? If not, would using the Spearman correlation be a better option?"*

We have used a criterion of 0.6 to eliminate highly correlated variables. We have included a reference to Ahn and Palmer (2016) who use criterion of 0.7 (lines 232-233). Our choice reflects a compromise between the more rigorous threshold of Ahn and Palmer (2016) and the suggested (0.5).

*"L327-331 stated that WS_Area and Stream_Length are removed whereas WS_Perimeter and Basin_Compactness are retain due to the intension to focus on basin shape rather than basin size for concentrating on the efficiency of the fluvial system transport. However, it should be aware that the length-area relationship has been found as a function of basin shape (Sassolas-Serrayet et al., 2018). Additionally, several studies have shown that basin size plays a critical role in catchment hydrological response and determines the curve of the flood frequency analysis (Merz & Blöschl, 2009). While it is recognized that the drainage size appears in the final model based on its origin as a one-variable model, the description provided by the authors may be misleading."*

The length-area relationship described by Sassolas-Serrayet et al. (2018) is included in this study as the Drainage_Density variable which is retained in the 19-variable regression. With regard to the basin size, we have provided clarification that the eliminated WS_Area variable describes total basin size and that we have

retained the contributing drainage area as it is more proportionally relevant to channel discharge (lines345-352). We thank Reviewer #1 for the additional references.

*"It is unclear why drainage size still appears in the 28 variables if it has already been recognized as a prominent variable in previous RFFA."*

The goal of this study was to improve upon the predictive strength of drainage area in a heterogeneous landscape. The predictive drainage area approach as a proxy for channel discharge is widely used by the geomorphic and earth science community. We explored variables that may improve upon its predictive strength of this relationship. This potentially avoids having to develop separate $Q$ vs $A_d$ relations for separate land use conditions. We feel the improved prediction (by approximately 5% across a broad range of flood quantiles) when including a second variable, the percentage of vegetated (naturalized) upstream area, will be of interest to members of the geomorphic earth science community, particularly those interested in estimating discharge in a heterogeneous landscape. For clarity and comprehension, we have more explicitly stated the goal of this study and the exploratory nature if this research (lines 86-90).

*"The link between the interpretation of the PCA results and Figure 6 needs improvement, and it is unclear what procedure/standard the authors used to objectively eliminate variables based on the results of the Pearson correlations and PCA."*

We have revised the interpretation of the PCA results to include the standards used to retain the first three dimensions of the analysis (lines 319-323).

*"the observed correlation" in L326 should be stated in a quantitative way."*

The correlation coefficients between Gradient_Mean and Gradient_StDev ($r$=-0.795) and between Mean_Rainfall and Mean Precipitation ($r$=0.697) have been included.

*"It is stated in L227-233 and L428-436 that the results of the best-fit distributions show an extreme low percentage for GEV distributions and the authors refer (or hint?) this to the number of parameters. Although it is not incorrect, it should be noted that the two two-parameter distributions used in this study (EV1 and LN) have lighter-tailed behavior than the two three-parameter distributions (GEV and LP3) (El Adlouni et al., 2008). Based on their results, it implies that the estimates of the extreme flood quantiles are relatively small in this region. Such a result may be due to the shorter dataset used in this study, which should be roughly between 37-48 years long based on the informative mean and standard deviation of the record length provided in this paper. As a result, there may be considerable uncertainty in estimating extreme quantiles for GEV (e.g., Papalexiou and Koutsoyiannis, 2013), which should be properly referenced in the discussion."*

We thank Reviewer #1 for the additional references. We have included an interpretation of tail behaviour in the Discussion (lines 462-467).

*"The title suggests a potential linkage or interaction between these conditions and flood frequency analysis, so it is expected that more discussion of this topic would be included (which is still weak, e.g., 450-452, and supporting references are expected). Otherwise, it may be advisable to reconsider the title to avoid any potential for confusion.*

*In L463-464, the authors imply that they intend to implement another modified model to properly capture spatial variability in a glacial region, but the resulting model obtained in this study eventually excluded all these variables. Please comment on this, and it is not clear whether the scope of this study is "glacial conditions" or "land use" or both from the current title."*

To emphasize the importance of different variables as flood frequency shifts from high frequency, low magnitude flood events to more extreme, low frequency overbank events, we have added Table 5 to summarize five- and three-variable models during backward elimination and support discussion (lines 491-509). In

addition, we have revised the title to be more explicit regarding the exploration of both glacial conditions and land use.

*"Please provide more information on the dataset used in Sec. 3.3, such as whether the same basin boundary was used for the analyses of the one-variable model and the two-variable model, the duration of streamflow data, and whether the analysis is representative if the duration of each variable does not overlap (as shown, the precipitation data is from 1981-2000, whereas the land use data is based on the dataset from 2014-2017)."*

The Methods and Data Collection section has been revised to more explicitly state that drainage area is is derived for each gauge station and employed for the single-variable RFFAs and the multi-variable RFFAs (lines 186-193).

*"L160-161: Since the authors explain the reason for using annual maxima instead of partial duration series (PDS), it would be appropriate to acknowledge other commonly used metrics, e.g., the peak-over-threshold (POT) analysis."*

A comment has been added to include the partial duration series (peak-over-threshold) approach (167-168).

*"There is no clear discussion in the previous section concerning the statement in L537-538, which appears as the closing sentence of the conclusion."*

The reference to width-to-depth ratios has been removed as it is outside the scope of this study.

*"It could be beneficial for the authors to consider providing some additional discussion on how the findings of this study may be transferable to other regions. Do the authors suggest directly including naturalized variables in RFFA for other regions, or do they recommend conducting an entire eliminating process based on different environmental variables obtained in the study area?"*

A brief discussion regarding transferability of these findings and model recalibration has been added (lines 527-530).

Throughout:

*"There appear to be a relatively large number of minor mistakes, such as unclear meanings of italics and special font sizes (e.g., L83, L86, L93, L310-312, L365, L366), using abbreviations before their definitions (e.g., L16, L87), mistakes in figures (e.g., the legend of Fig. 2: HYDAT is not a GIS dataset; the y-ticks of Fig. 7 are missing; Dim1 explains 27.7% in L318 but 28% in Fig. 6), and typos and grammatical errors. It may be beneficial to have someone do a thorough proofreading to address these issues.*

*It is suggested that several statements should provide supporting references (e.g., L155, L225 (n/p<40), L469-471)."*

All minor editorial corrections have been made.

**Reviewer #2**

Major comments:

1) *"In my opinion, the novelty aspect is not strong enough for a publication in HESS. The methods used are not new, and the results are expected: the catchment area is the main explanatory factor, and all other catchment attributes are far behind. However, with some improvements, this paper could be resubmitted to other journals in Geography (for the GIS treatment aspects) or in Hydrology (but as a regional study focusing on glacially influenced catchments)."*

We thank Reviewer #2 for their valuable comments; however, we disagree regarding the strength of interest in the findings of our paper. The widely used drainage area to discharge relationship is frequently applied by the geomorphic and earth science community as well as watershed planners and we feel our paper will be of significant interest to those doing rapid geomorphic assessments. Our findings highlight the central importance of considering the appropriate flood frequency distribution and channel forming discharges (not considered by hydrologists). The goal of this study was to improve upon the predictive strength of drainage area in a heterogeneous landscape. We explored variables that may improve upon the predictive strength of this relationship. We feel the improved prediction (by approximately 5% across a broad range of flood quantiles) when including a second variable, the percentage of vegetated (naturalized) upstream area, will be of interest to members of the geomorphic earth science community, particularly those interested in estimating discharge in a heterogeneous landscape. For clarity and comprehension, we have more explicitly stated the goal of this study and the exploratory nature of this research (lines 86-90) and we have restructured the introduction to make the RFFA discussion more prominent (lines 35-49).

2) *"Some important references in RFFA are missing. Typically, RFFA (which consists of transferring information available at different gauges to a target station) is used to achieve two different objectives. The first is to improve the estimation of flood quantiles at the target station for high return periods. The second is to estimate quantiles when no discharge data are available (ungauged target station). Neither of these issues are addressed in this paper. High return period quantiles (up to 100 years) are estimated locally (sometimes with only a few years of observation), and the results of the models obtained are not tested for "ungauged" cases. To do this, a leave-one-out (LOO) cross-validation methodology would have been required."*

A leave-one-out cross validation has been included that supports our findings. The results have been included in Table 7.

Minor comments:

*"L160: This statement is not true. I don't think Cunnane said it. There is a lot of literature explaining the interest of partial duration series (or peaks over threshold)."*

This statement has been revised to acknowledge the gained efficiency in using the AMS approach for floods $Q(T)$ when $T>10$ (lines 168-169)

*"L171: a large error is associated with a 100-year quantile."*

We have acknowledge the increased uncertainty in predicting discharge for larger, less frequent events and the limited length of gauge records (lines 296-298; 412-416; 465-467).

*"L186: your 28 attributes should be presented in a data section at the beginning of the paper (not in table 3, in the results section)."*

We have revised our paper to include a Regression Model Input section that includes computation of flood quantiles (dependent variables) and the parameterization of the 28 variables (lines 242-292).

*"L188: I did not understand these "sub-basin" characteristics."*

For clarity and reproducibility, we have revised the methods section with section 3.2.2 dedicated to the characterization of sub-basin attributes (lines 194-218).

*"L218-227: This is not a result. It should be moved to the methods part."*

We agree with Reviewer #2 that these lines should be moved to methods. This information has been moved to Sub-basin attributes (lines 195-198) and Estimation of flood frequency (lines 160-162 and lines 172-179).

*"L278-288: this is not a result. This should be moved to the data part (even: figure 5). L300-305: this is not a result. This should be moved to the data part (also table 3)."*

This information has been moved to the data section (lines 272-292).

*"L350-360: please show the results that lead to this conclusion."*

A summary of the results of the 19-variable regression has been added to the Results section (Table 4)

*"L469-471: please show the results that lead to this conclusion."*

A summary of the variables retained in the five- and three-predictor models has been added to the Results section (Table 5; lines 391-397) to emphasize the importance of different variables as flood frequency shifts from high frequency, low magnitude flood events to more extreme, low frequency overbank events (lines 495-509).

Again, we thank the reviewers for their valuable input and have made significant revisions to our paper to address all the issues that have been raised.

---

## Author Response (AR2)

We thank the reviewers for the insights they have provided regarding our research paper, "Modelling flood frequency and magnitude in a glacially conditioned, heterogeneous landscape: testing the importance of land cover and land use".

References to line numbers below are to the corrected manuscript.

**Reviewer #1**

Major comments:

*"L218-220: It not clear whether "sub-watersheds" and "sub-basin units" refer to the same concept or different concepts."* there is still one (at least) being missed here (lines 195-200)

All remaining references to "sub-watersheds" has been corrected to read "sub-basins" (L196). Sub-basin only is now used throughout the manuscript.

Minor comments:

*"Concerning the Pearson correlation: …. (2) Does the normality been checked? If not, would using the Spearman correlation be a better option?"* the authors did not reply to this question!?

Following Tacq (2010), multi-variate tests for normality have not been performed (noted L361-365) since examination of residuals was conducted and is a streamlined way of addressing this issue. The important drainage area variable, however, was tested and transformed for inclusion in the single-variate model. The analysis was repeated using Spearman rank correlations and the results agree with the Pearson correlation tests. The manuscript has been modified to include the results of the Spearman correlation tests. (L335-359)

Throughout:

*"There appear to be a relatively large number of minor mistakes, such as unclear meanings of italics and special font sizes (e.g., L83, L86, L93, L310-312, L365, L366), using abbreviations before their definitions (e.g., L16, L87), mistakes in figures (e.g., the legend of Fig. 2: HYDAT is not a GIS dataset; the y-ticks of Fig. 7 are missing; Dim1 explains 27.7% in L318 but 28% in Fig. 6), and typos and grammatical errors. It may be beneficial to have someone do a thorough proofreading to address these issues.*

*It is suggested that several statements should provide supporting references (e.g., L155, L225 (n/p<40), L469-471)."* Hard to check because the corresponding new lines are not indicated by the authors

All minor editorial corrections have been made.

E.g., abbreviations removed (L17, L93); italics and font reformatted (L89, L92, L99, L335-339, L383, L384); Fig. 2 legend revised to georeferenced datasets; Fig. 7 y-ticks inserted; Fig. 7 Dim1 to 28.0%; supporting references provided (L166, (n/p<40) L177, Table 5 results provided L390).